# Immunoglobulin expression in the endoplasmic reticulum shapes the metabolic fitness of B lymphocytes

Huda Jumaa[1,2], Marieta Caganova[3] ◉, Ellen J McAllister[1,2], Laura Hoenig[4], Xiaocui He[2], Deniz Saltukoglu[1,2] ◉,
Kathrin Brenker[2], Markus Köhler[8], Ruth Leben[7,8], Anja E Hauser[5,6], Raluca Niesner[7,8], Klaus Rajewsky[3], Michael Reth[1,2],
Julia Jellusova[1,2] ◉

**The major function of B lymphocytes is to sense antigens and to produce protective antibodies after activation. This function requires the expression of a B-cell antigen receptor (BCR), and evolutionary conserved mechanisms seem to exist that ensure that B cells without a BCR do not develop nor survive in the periphery. Here, we show that the loss of BCR expression on Burkitt lymphoma cells leads to decreased mitochondrial function and impaired metabolic flexibility. Strikingly, this phenotype does not result from the absence of a classical Syk-dependent BCR signal but rather from compromised ER expansion. We show that the reexpression of immunoglobulins (Ig) in the absence of the BCR signaling subunits Igα and Igβ rescues the observed metabolic defects. We demonstrate that immunoglobulin expression is needed to maintain ER homeostasis not only in lymphoma cells but also in resting B cells. Our study provides evidence that the expression of BCR components, which is sensed in the ER and shapes mitochondrial function, represents a novel mechanism of metabolic control in B cells.**

## Introduction

The primary role of the BCR on mature B cells is to recognize antigen and to initiate a signaling cascade resulting in cell activation and clonal selection. The BCR is assembled inside the ER from four components, namely, membrane-bound Ig (mIg) heavy (H) chain, light (L) chain, and the signaling subunits Igα and Igβ (CD79a and CD79b), a process that is required for the transport and deposition of the BCR on the cell surface (Reth & Wienands, 1997; Gold & Reth, 2019). The ER is not only the site of protein synthesis and folding but also can contribute to the regulation of cellular metabolism. ER-

associated proteins such as BiP, XBP1, or PERK have been shown to regulate protein synthesis and lipid metabolism (Bravo et al, 2013). In addition, the ER plays a crucial role in calcium homeostasis and can alter mitochondrial function by exchanging ions and other molecules through ER–mitochondrial contact sites (Tubbs & Rieusset, 2017).

In the resting state, the BCR forms oligomers (Yang & Reth, 2010), which are opened upon antigen binding allowing Src family kinases such as Lyn and the spleen tyrosine kinase (Syk) to interact with the immunoreceptor tyrosine-based activation motifs (ITAMs) of Igα and Igβ. Syk plays an essential role in signal initiation and amplification upon BCR engagement, and Syk-deficient B cells display severe functional defects and impaired survival (Turner et al, 1995; Klasener et al, 2014). BCR stimulation on mature B cells leads to an increase in cell mass and metabolic reprogramming as cells prepare for proliferation (Caro-Maldonado et al, 2014). In addition to playing a central role in B-cell activation, the BCR has also been shown to support survival of naïve mature B cells. B cells that because of a defective H or Igα gene are BCR negative display a reduced survival, demonstrating the importance of the BCR in B cell maintenance (Lam et al, 1997; Kraus et al, 2004).

Most B-cell lymphomas maintain BCR expression and are implicated to use BCR-signaling processes for their continuous activation (Niemann & Wiestner, 2013; Young et al, 2015; Burger & Wiestner, 2018). BCR-deficient lymphoma cells display a competitive disadvantage in comparison with wild-type lymphoma cells (Varano et al, 2017; He et al, 2018). Malignant B cells are characterized by increased metabolic activity to support their high proliferation. Oncogenic signaling frequently involves aberrant activation of metabolic regulators such as PI3K, mTOR, or cMyc to enhance nutrient acquisition and utilization (Franchina et al, 2018). The role of the BCR in regulating cell metabolism in lymphoma cells is currently poorly understood.

Here, we provide novel insight into BCR-dependent metabolic regulation in lymphoma cells. We show that B lymphoma cells with

---

[1]BIOSS Centre for Biological Signalling Studies and Centre For Integrative Biological Signalling Studies, Albert Ludwigs University of Freiburg, Freiburg, Germany [2]Department of Molecular Immunology, Institute of Biology III at the Faculty of Biology, Albert Ludwigs University of Freiburg, Freiburg, Germany [3]Max Delbrück Center for Molecular Medicine in the Helmholtz Association, Berlin-Buch, Germany [4]Ruhr-University Bochum, Bochum, Germany [5]Immune Dynamics Deutsches Rheuma-Forschungszentrum, A Leibniz Institute, Berlin, Germany [6]Immune Dynamics, Rheumatology and Clinical Immunology, Charité–Universitätsmedizin, Berlin, Corporate Member of Freie Universität Berlin, Humboldt-Universität zu Berlin, and Berlin Institute of Health, Berlin, Germany [7]Dynamic and Functional In Vivo Imaging, Veterinary Medicine, Freie Universität Berlin, Berlin, Germany [8]Biophysical Analytics, Deutsches Rheuma-Forschungszentrum, A Leibniz Institute, Berlin, Germany

Correspondence: julia.jellusova@bioss.uni-freiburg.de
Xiaocui He's present address is La Jolla Institute for Immunology, La Jolla, CA, USA

---

a defective BCR expression fail to expand their ER, which is accompanied by impaired mitochondrial function and other metabolic defects. This defect is rescued by Ig expression and does not require the production of a signaling-competent BCR. Moreover, we find the maintenance of ER mass to be coupled to Ig expression in naïve B cells as well, suggesting that the role of the BCR in governing ER homeostasis is not limited to lymphoma cells.

## Results

### BCR expression boosts B lymphoma fitness but is not absolutely required for survival

To analyze the role of the BCR in regulating metabolic activity, we rendered the human Burkitt lymphoma cell line Ramos deficient for all four BCR components (mIg H, L, Igα, and Igβ), here referred to as

BCR-KO (Fig 1A and B). Consistent with previous reports showing that BCR ablation does not lead to cell death in cMyc-driven lymphoma (Varano et al, 2017; He et al, 2018), we did not observe any significant differences in cell viability and proliferation between BCR-KO and wild-type (WT) cells (Fig 1C). However, these cells expressed lower levels of hexokinase II (HK-II), the enzyme mediating the first step of glycolysis, and a decreased S6 phosphorylation (Fig 1D) indicative of reduced mTORC1 activity. Consistent with reduced mTORC1 signaling, BCR-KO cells were more susceptible to cell growth arrest induced by the mTORC1 inhibitor rapamycin (Fig S1A). As mTORC1 is a central regulator of protein synthesis (Albert & Hall, 2015; Saxton & Sabatini, 2017), we assessed cell mass and total protein levels of BCR-KO and WT Ramos cells but did not observe any significant differences (Figs 1E and S1B). To confirm our findings that lymphoma cells are able to adapt to the loss of the BCR, we also studied another BCR-negative Burkitt lymphoma line, namely, DG75 with a defective mIg H gene (herein referred to as H-KO cells, Fig 1F). The H-KO cells were significantly

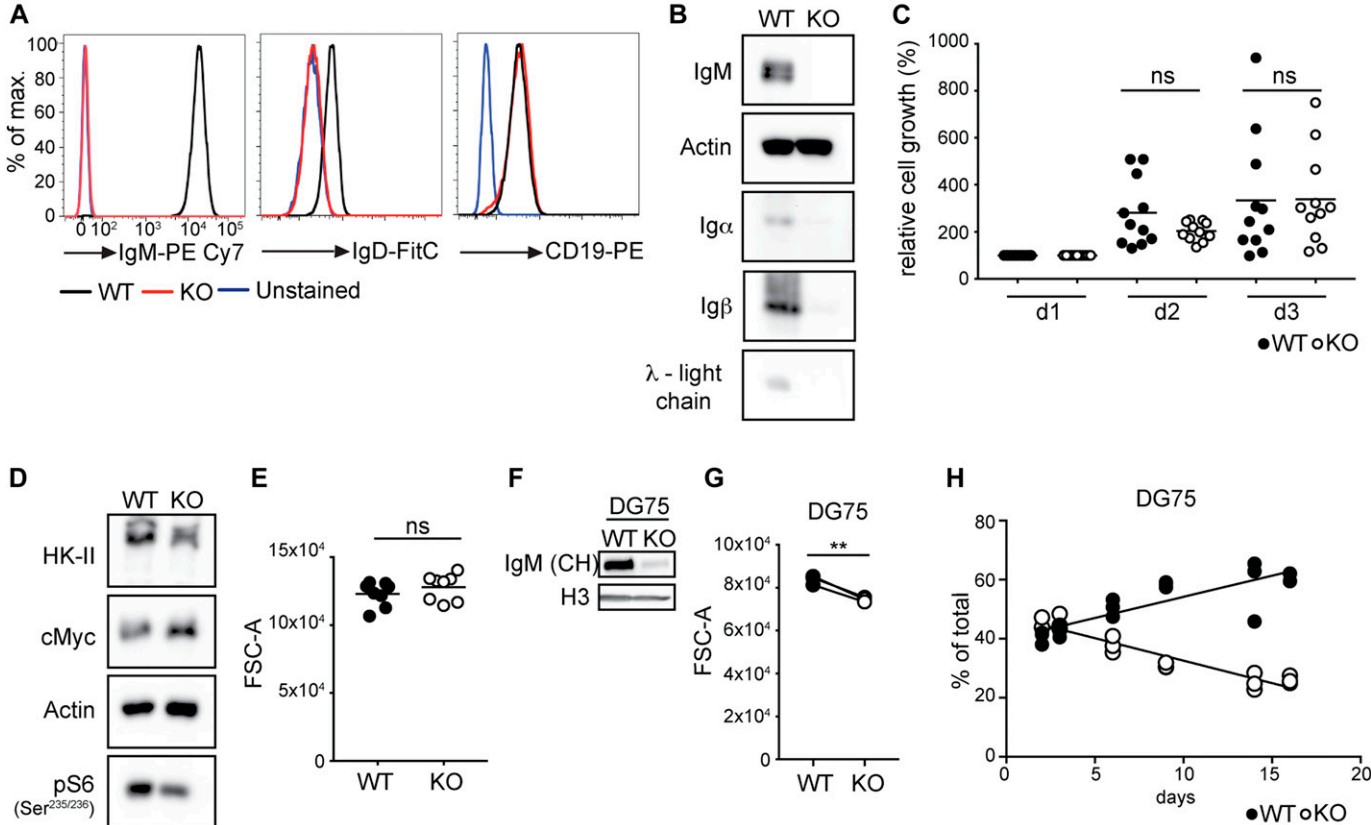

**Figure 1. BCR expression boosts B lymphoma fitness but is not absolutely required for survival.**
**(A)** Cells were stained with anti-IgM and anti-IgD to examine cell surface expression of the BCR. Anti-CD19 was used as a control. **(B)** Expression of the μ-heavy chain, λ-light chain, Igα, and Igβ were determined by Western blot. One of three independent experiments is shown. **(C)** The cells were plated on d0 and cell numbers were assessed on d1, d2, and d3 using the CCK8-kit. Values were normalized to the measurement obtained on d1. Significance was determined using the ANOVA test. **(D)** Expression of the indicated proteins was determined by Western blot. One of four independent experiments is shown. **(E)** Forward scatter (FSC-A) as a measure of cell size was determined using flow cytometry. Significance was determined using the Mann–Whitney test. N = 8. **(F)** Biological triplicates of Ig heavy chain–deficient DG75 cells were sorted on d3 upon Ig heavy chain deletion and pooled for analysis. Expression of the Ig heavy chain (antibody directed against the CH domain) and histone 3 (H3) was determined by Western blot. One of two independent experiments is shown. **(G)** Cell size measured on d6 after Ig heavy chain deletion. Significance was determined using the paired *t* test. N = 3; **$P$ = 0.0074. **(H)** Relative abundance of WT and H-KO DG75 cells in a mixed culture at the indicated time points after Ig heavy chain deletion. Linear regression analysis was performed. Slopes of the WT and KO abundance lines were found to be significantly different. $P$ < 0.0001. Circles represent results from independent experiments. **(A, B, C, D, E, F, G, H)** KO, BCR-KO Ramos cells (A, B, C, D, E), or H-KO DG75 cells (F, G, H) WT, wild-type cells of the corresponding cell line.

smaller than the WT DG75 cells (Fig 1G) and less competitive in a co-culture with WT cells but were able to proliferate and survive for an extended period of time (Fig 1H). Taken together, these results show that although BCR ablation does not prevent proliferation, these cells display reduced fitness.

## Metabolic activity is reduced in the absence of the BCR

Given the decreased mTORC1 activation and HK-II expression in BCR-KO Ramos cells, we hypothesized that their metabolic activity is altered. We, thus, compared the metabolome of BCR-KO and WT Ramos cells and found 11 metabolites to be significantly increased and 123 metabolites to be significantly decreased in BCR-KO cells. Particularly, significant changes of the abundance of the intermediates of glutamine and lipid metabolism (Fig S2A) suggest that the absence of the BCR is accompanied by extensive metabolic reprogramming. NADH and NAD(P)H serve as cofactors for various enzymes governing energy production, biosynthesis, and redox balance and are converted to their oxidized forms NAD+ and NADP+, respectively (Blacker & Duchen, 2016). To further investigate the metabolic program of BCR-KO Ramos cells, we performed NAD(P)H fluorescence lifetime microscopy. In this approach, NAD(P)H autofluorescence is used to assess the relative abundance of free (short fluorescence lifetime [450 ps] [Lakowicz et al, 1992]) versus enzyme-bound NAD(P)H (longer fluorescence lifetime). Upon mitochondrial electron transport chain (ETC) inhibition, the free versus bound NAD(P)H balance shifts towards free NAD(P)H (Stringari et al, 2012; Blacker & Duchen, 2016). We confirmed this observation in WT Ramos cells. Treatment with the ETC inhibitor rotenone resulted in increased free NAD(P)H (Fig S2B). On the other hand, glycolysis inhibition leads to a decreased bound-unbound NAD(P)H ratio (Stringari et al, 2012; Blacker & Duchen, 2016). Thus, the assessment of free versus Enzyme-bound NAD(P)H allows the interrogation of the metabolic state on a single-cell level and provides information on the relative usage of the two main energy-generating pathways: glycolysis and mitochondrial respiration. We found NAD(P)H fluorescence lifetime to be decreased in BCR-KO Ramos cells in comparison with WT cells, suggesting relatively higher levels of free NAD(P)H in BCR-KO Ramos cells (Figs 2A–C and S2C), which was confirmed in our metabolomic study by slightly lower NAD+ levels in BCR-KO Ramos cells (Fig S2D). The shift in the phasor plot observed in the BCR-KO cells is not only indicative of relatively decreased NAD(P)H binding but also suggests that the remaining enzymes binding NAD(P)H in BCR-KO cells are different from those binding NAD(P)H in WT cells (Figs 2C and S2C). In summary, the obtained results are indicative of an overall reduced metabolic activity in BCR-KO Ramos cells, which is likely associated with a metabolic shift away from oxidative phosphorylation towards glycolysis.

## Glycolytic flux is not affected by the absence of the BCR

To test whether the absence of the BCR affects glucose acquisition, we measured fluorescent glucose analogue 2-(N-(7-nitrobenz-2-oxa-1,3-diazol-4-yl)amino)-2-deoxyglucose (2-NBDG) uptake after 5 and 30 min and calculated the rate of glucose consumption over

time. We found glucose uptake to be significantly higher in BCR-KO Ramos cells than in WT cells (Fig 2D).

To assess the rate of glycolysis in WT and BCR-KO Ramos cells, we measured the extracellular acidification rate (ECAR) in real-time using Seahorse flux technology. Addition of glucose caused an increase in ECAR in both WT and BCR-KO Ramos cells (Fig 2E). No significant differences in the glycolytic rate or the glycolytic capacity were observed between WT and BCR-KO Ramos cells (Fig 2E and F). To verify that glucose metabolism is essential for the survival of Ramos cells, we treated the cells with 2-deoxy-D-glucose (2DG) to inhibit glycolysis and assessed cell numbers after 1 and 2 d. Both WT and BCR-KO Ramos cells treated with 2DG showed a comparable and significant decrease in proliferation (Fig 2G). In conclusion, glycolytic flux is not affected by the loss of the BCR in Ramos cells.

## BCR ablation leads to impaired mitochondrial function

Because glycolysis was not affected in BCR-KO Ramos cells, a shift towards free NAD(P)H could be caused by impaired respiration. To analyze mitochondrial respiration in BCR-KO and WT Ramos cells, we measured oxygen consumption in cells that were provided with glucose, glutamine, and pyruvate. Subsequently, the cells were treated with oligomycin, an inhibitor of mitochondrial ATPase, the uncoupling agent carbonyl cyanide 4-(trifluoromethoxy) phenyl-hydrazone (FCCP) and inhibitors of mitochondrial complex I and III (rotenone and antimycin). BCR-KO Ramos cells showed slightly decreased basal respiration and significantly decreased maximal oxygen consumption and spare respiratory capacity (Fig 3A and B). Similarly, H-KO DG75 cells showed reduced spare respiratory capacity (Fig 3C). The mitochondrial function in BCR-KO cells could be impaired either because of reduced activity of the respiratory chain or because of cytosolic regulation of mitochondrial processes. To differentiate between these two scenarios, we analyzed activity of the respiratory complex I in BCR-KO Ramos cells. To this end, we permeabilized the cells and provided them with the complex I substrates pyruvate and malate; ADP was added in a second step. State 3$^{oligo}$ respiration, a readout of complex I activity and defined as the difference between oxygen consumption with the substrates pyruvate, malate, and ADP and oxygen consumption after oligomycin treatment, was significantly reduced in BCR-KO cells (Fig 3D and E). These results suggest that complex I function is reduced in BCR-KO cells. In addition to oxygen consumption, we measured accumulation of reactive oxygen species (ROS) in BCR-KO and WT Ramos cells. ROS is a natural by-product of mitochondrial function but can also be produced in the ER or cytoplasm (Chandel, 2017). We used the probe H2CFDA, which becomes fluorescent upon $H_2O_2$ exposure to measure ROS production in BCR-KO and WT Ramos cells. Cells treated with the mitochondrial inhibitor oligomycin served as a negative control for H2CFDA staining (Fig 3F). We calculated the difference between values obtained from untreated and oligomycin-treated cells and found ROS production to be significantly decreased in BCR-KO Ramos cells (Fig 3F and G). As mTOR signaling is known to drive mitochondrial biogenesis (Morita et al, 2013) and mTOR signaling is reduced in BCR-KO Ramos cells, we hypothesized that reduced mitochondrial activity in BCR-KO Ramos cells may stem from inhibited mitochondrial biogenesis. To

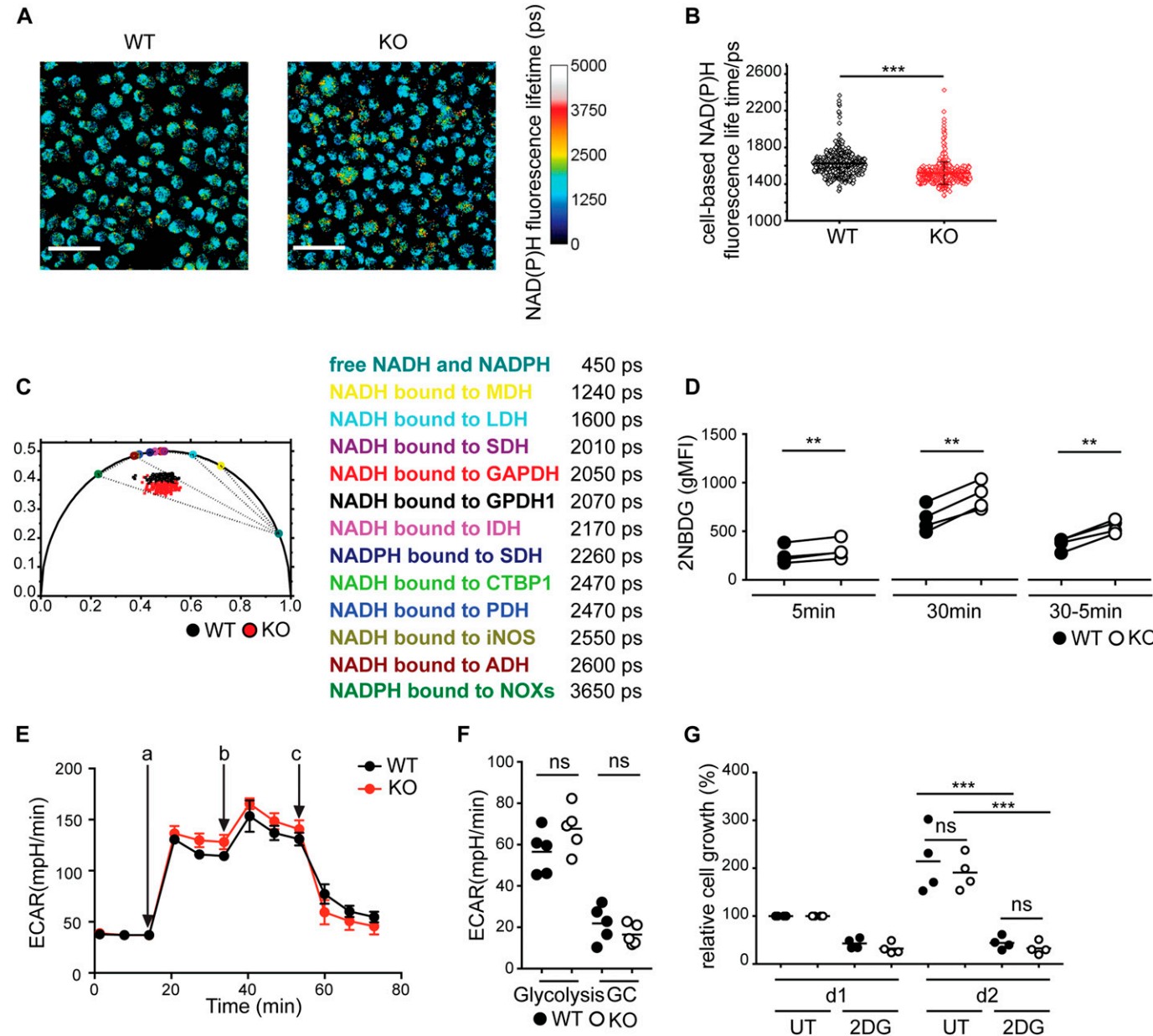

**Figure 2. Glycolytic flux is not affected by the absence of the BCR.**
**(A)** NAD(P)H fluorescence lifetime images of WT and KO cells, representative for two experiments. Color gradient indicates NAD(P)H fluorescence lifetime (ps). Scale bar indicates 50 μm. **(B)** The NAD(P)H fluorescence lifetimes averaged over single cells. Significance was determined using the Mann–Whitney test. ***$P$ < 0.001. The error bars represent SD values. Results representative for two independent experiments. **(C)** Cell-based phasor plot of 941 KO cells and 526 WT cells out of two independent experiments is shown. For reference, color-encoded phasor coordinates representative for pure free NAD(P)H and for NAD(P)H bound to various pure enzymes are depicted on the half circle (the color coding is explained in the table, together with the corresponding fluorescence lifetimes). Dotted lines represent possible combinations of free and enzyme-bound NAD(P)H. **(D)** Cells were incubated with 2NBDG for 5 and 30 min. Glucose uptake was determined by flow cytometry. The difference between 30 and 5 min was calculated to determine the rate of glucose uptake over time. Paired two-tailed $t$ test was used to determine statistical significance. N = 4; **$P$ = 0.002, 0.0034, and 0.0029; gMFI, geometric mean fluorescence intensity. **(E)** Cells were resuspended in glucose-free medium containing glutamine. ECAR was measured using Seahorse flux technology. The measurement was performed in technical triplicates and is displayed as mean ± SD. One of five independent experiments is shown. a, glucose; b, oligomycin; c, 2DG. **(E, F)** Summary of the experiments performed in (E). Unpaired $t$ test was used to determine significance. GC, glycolytic capacity. **(G)** Cells were plated on d0 with or without 2DG and cell numbers were assessed on d1 and d2 using the CCK8-kit. Values were normalized to the measurement obtained on d1. Differences observed on d2 were tested for significance using the ANOVA test. N = 4; ***$P$ = 0.0003 and 0.0005. Circles represent independent experiments. **(E)** For experiments in which technical replicates were measured in parallel (E), a circle represents the mean value of these replicates. KO, BCR-KO Ramos cells; WT, Ramos wild-type.

test this hypothesis, we analyzed mitochondrial mass by flow cytometry and microscopy and found no significant differences between BCR-KO and WT Ramos cells (Figs 3H and S3A), suggesting that impaired mitochondrial function is not derived from decreased mitochondrial biogenesis in BCR-KO lymphoma cells.

## The BCR supports metabolic flexibility

Our data demonstrate that despite decreased mTORC1 signaling, BCR-KO Ramos cells are able to proliferate and survive. We hypothesized that although these cells are able to survive under normal cell culture conditions, they might be more sensitive to metabolic stress. Reduced ATP levels are known to result in 5' adenosine monophosphate-activated protein kinase (AMPK) activation and subsequently mTOR inhibition to prevent further cell growth and associated energy expenditure (Albert & Hall, 2015; Saxton & Sabatini, 2017). To inhibit mitochondrial ATP generation, we treated the cells with oligomycin and analyzed growth and survival. Oligomycin inhibits mitochondrial ATPase; however, it does not directly affect ATP generation from glycolysis. Indeed, we found that unlike WT the BCR-KO Ramos cells did not expand in the presence of 1 $\mu$M oligomycin (Fig S3B). To test whether mitochondrial ATP was needed for the survival of the cells, we measured cell death by 7-aminoactinomycin D (7-AAD) incorporation after overnight treatment with oligomycin (Fig 3I and J). Although the survival of WT Ramos cells was not significantly affected by oligomycin treatment, ~30% of the BCR-KO Ramos cells showed 7-AAD incorporation, indicating increased cell death (Fig 3I and J). To verify that oligomycin retains its inhibitory function in an overnight culture, we measured oxygen consumption of oligomycin-treated cells. We found that oligomycin treatment resulted in an efficient inhibition of mitochondrial ATPase after overnight incubation in both WT and BCR-KO Ramos cells (Fig S3C), demonstrating that oligomycin remains active throughout the overnight treatment. In addition, we measured cell survival with oligomycin in the medium where glucose was replaced by other carbon donors such as galactose or GlutaMAX. To gain energy, both galactose and GlutaMAX need to be catabolized in the mitochondria. In contrast, if glucose is available, ATP can be generated without the need for the mitochondrial respiratory chain. As expected, oligomycin completely inhibited cell survival in both WT and BCR-KO Ramos cells if the cells were provided with galactose or GlutaMAX (Fig S3D). In the presence of glucose, however, the survival of BCR-KO but not WT Ramos cells was reduced by oligomycin (Fig S3D). Furthermore, metabolic profiling of oligomycin-treated BCR-KO and WT cells revealed that whereas intracellular glucose levels were significantly higher in BCR-KO Ramos cells in comparison with WT cells, the levels of glucose-6-phosphate (G6P) were significantly lower (Fig 3K). G6P is a metabolite produced from glucose through hexokinase activity; thus, these results are consistent with the observed reduction in HK-II expression in BCR-KO Ramos cells. Combined, these data suggest that upon mitochondrial inhibition, BCR-KO Ramos cells are unable to solely rely on glycolytic ATP generation.

## Syk-dependent BCR signaling is dispensable for metabolic adaptations in B lymphoma

The tyrosine kinase Syk is an essential component of the BCR signaling pathway. To test whether or not the observed metabolic defect in BCR-KO cell is due to defective BCR signaling, we generated Syk-deficient (Syk-KO) Ramos B cells (Fig S4A). Surprisingly, Syk-KO cells do not display a metabolic defect as indicated by normal S6 phosphorylation and HK-II expression in comparison with WT cells (Fig 4A). Furthermore, oxygen consumption and ROS production were comparable between Syk-KO and WT Ramos cells (Fig 4B and C). Moreover, unlike BCR-KO cells, Syk-KO Ramos cells showed normal growth and survival after oligomycin treatment suggesting that increased cell death observed in BCR-KO cells treated with oligomycin is not a result of impaired Syk-mediated signaling (Fig 4D). To verify that Syk-KO Ramos B cells are defective for BCR signaling, we exposed WT, Syk-KO, and BCR-KO Ramos to pervanadate and found that the latter two cells are unable to mobilize calcium (Fig 4E). This result is consistent with studies demonstrating that Syk is essential for pervanadate induced calcium flux (Patterson et al, 2015). The Syk-KO and BCR-KO Ramos cells also failed to phosphorylate SLP65 and Erk in response to anti-IgM stimulation (Fig S4B). In conclusion, Syk-KO Ramos cells show the expected BCR signaling defects but normal mitochondrial function. Thus, mitochondrial defects observed in BCR-KO lymphoma cells cannot be attributed to the loss of BCR-mediated signaling.

## ER expansion and function are impaired in BCR-KO B lymphoma cells

As Syk-mediated signaling was not responsible for the metabolic defects in BCR-KO Ramos cells, we hypothesized that a signaling-independent function of the BCR could mediate the observed phenotype. The BCR components are first expressed and assembled inside the ER and the proper function of this compartment depends on the Ig production (Iwakoshi et al, 2003). We, thus, tested whether the ER mass was altered in BCR-KO Ramos cells. For this, we stained WT and BCR-KO Ramos cells with ER-tracker green and found ER mass to be significantly decreased in the latter cells (Fig 5A–C). Next, we analyzed the expression of ER-associated proteins such as PERK and BiP. Expression of both of these proteins was strongly decreased in BCR-KO Ramos cells in comparison with WT cells (Fig 5D). ER stress induction using tunicamycin or thapsigargin led to increased BiP levels in both WT and BCR-KO Ramos cells; however, BiP levels remained lower in BCR-KO Ramos cells as compared with WT cells (Fig 5D). As a control, the cells were treated with the chemical chaperone 4-sodium phenyl butyrate. BiP levels were lower in BCR-KO Ramos cells than in WT cells and in both cell types, BiP levels after sodium phenyl butyrate treatment were comparable with untreated cells (Fig 5D). Similarly to BiP, spliced Xbp1 (sXbp1) was strongly increased upon thapsigargin treatment (Fig 5E). Although the sXbp1 protein was undetectable in BCR-KO cells, its abundance increased upon thapsigargin treatment but remained below the levels detected in WT Ramos cells (Fig 5E). Similar to Ramos cells, H-KO DG75 cells displayed reduced ER mass, BiP, and sXbp1 expression (Fig 5F–I), confirming that BCR expression is needed to maintain ER function and homeostasis in B lymphoma cells.

To measure signaling independent calcium efflux from the ER, we incubated WT, BCR-KO, and Syk-KO Ramos cells with EGTA, which removes extracellular calcium and treated the cells with

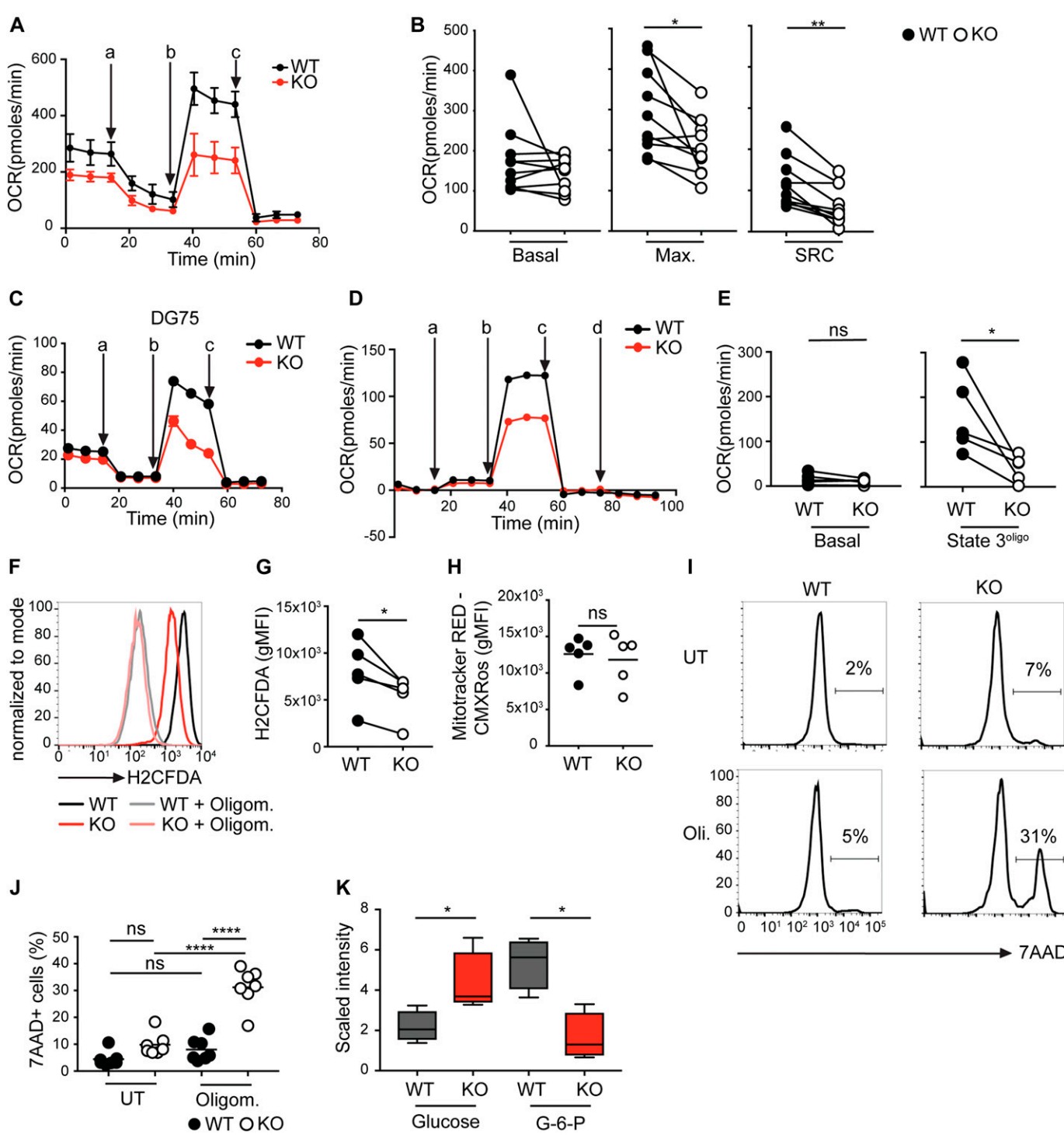

**Figure 3. BCR ablation leads to impaired mitochondrial function and reduced metabolic flexibility.**
**(A)** WT and BCR-KO Ramos cells were resuspended in medium containing pyruvate, glutamine, and glucose, and oxygen consumption rate (OCR) was measured using Seahorse flux technology. The measurement was performed in technical duplicates (WT) or triplicates (KO) and is displayed as mean ± SD. One of 10 experiments is shown. a, oligomycin; b, FCCP; c, rotenone + antimycin. **(A, B)** Summary of the experiments performed in (A). Paired two-tailed *t* test was used to determine significance. N = 10, *P = 0.0136, **P = 0.0011. Max, maximal oxygen consumption; SRC, spare respiratory capacity. **(C)** Oxygen consumption of WT and H-KO DG75 cells. **(A)** Experiments were performed as in (A). The experiment was performed in eight technical replicates and is displayed as mean ± SD. The experiment is representative for three independent experiments. **(D)** Cells were permeabilized with saponin, OCR was measured over time. a, pyruvate + malate; b, ADP; c, oligomycin; d, rotenone + antimycin. One of five independent experiments is shown. **(D, E)** Summary of experiments performed in (D). Significance was determined using a two-tailed paired *t* test. *P = 0.0352. **(F)** Untreated and oligomycin-treated cells were loaded with H2DCFDA and analyzed by flow cytometry. One of five independent experiments is shown. **(F, G)** Difference between untreated and oligomycin-treated cells stained for reactive oxygen species as shown in (F) was calculated. Paired two-tailed *t* test was used to determine

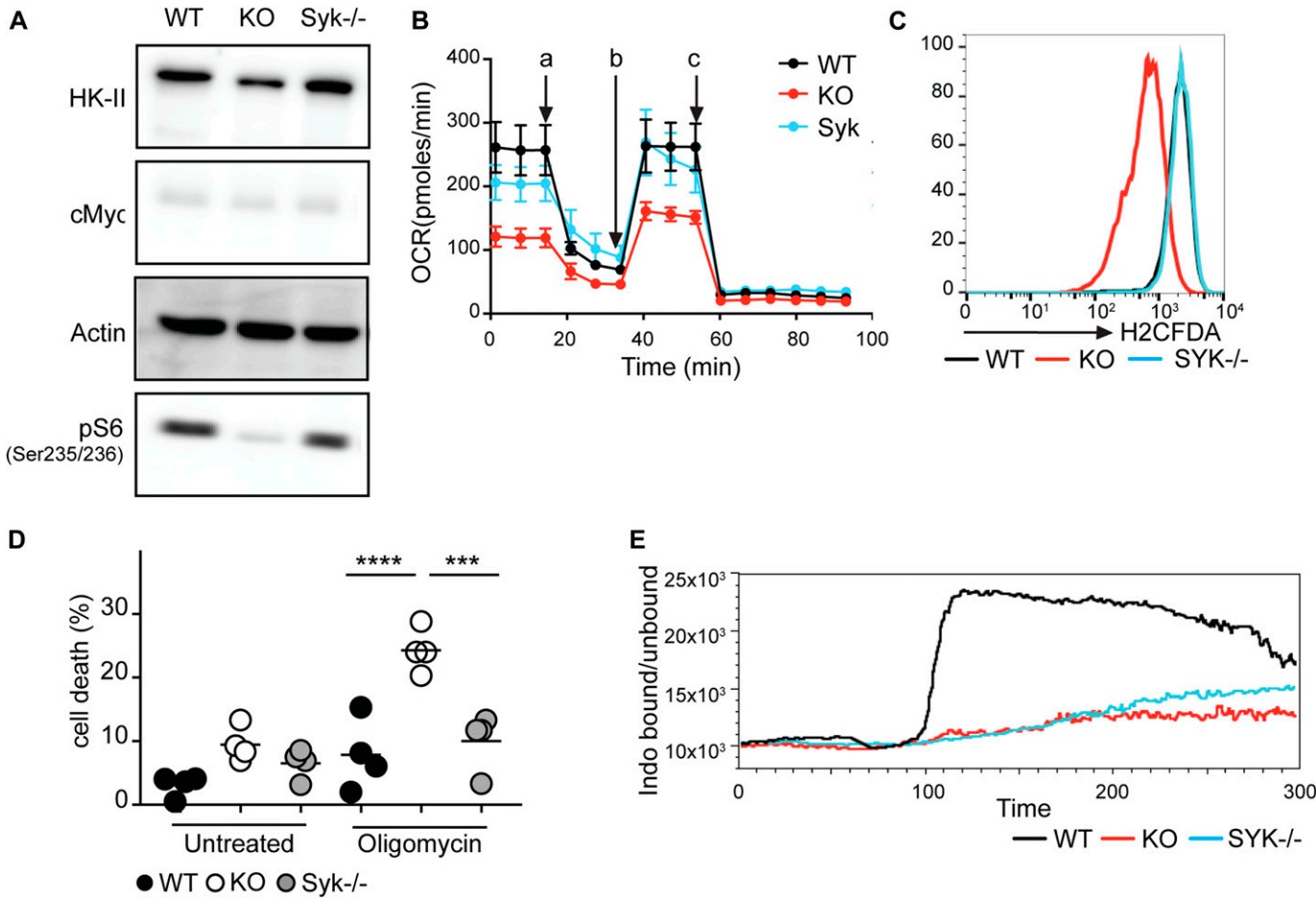

**Figure 4. Syk-dependent BCR signaling is dispensable for metabolic adaptations in lymphoma B cells.**
**(A)** Expression of the indicated proteins was determined by Western blot. Actin served as a loading control. One of three independent experiments is shown. **(B)** Cells were resuspended in medium containing pyruvate, glutamine, and glucose, and OCR was measured using Seahorse flux technology. The measurements were performed in technical triplicates and are displayed as mean ± SD. One out of two independent experiments is shown. a, oligomycin; b, FCCP; c, rotenone + antimycin. **(C)** Cells were stained with H2CFDA to detect reactive oxygen species. One of two independent experiments is shown. **(D)** Cells were cultured overnight with or without oligomycin. Cell death was determined using 7-AAD. Significance was determined using the ANOVA test. For clarity, only comparisons between KO and WT or Syk-KO cells are shown. N = 4, ***$P = 0.004$, ****$P < 0.0001$. **(E)** Cells were loaded with Indo-1 AM. Basal calcium levels were measured for 60 s. Cells were stimulated with pervanadate. Circles represent independent experiments. KO, BCR-KO Ramos cells; Syk-KO, Syk-deficient Ramos cells; WT, Ramos wild-type.

thapsigargin, an inhibitor of the ER-associated calcium pump. Calcium flux after thapsigargin treatment was decreased in BCR-KO Ramos cells in comparison with WT Ramos cells (Fig 5J). Notably, calcium flux was comparable between Syk-deficient and WT Ramos cells (Fig 5J), demonstrating that thapsigargin-induced calcium mobilization is Syk independent. In summary, the absence of the BCR results in decreased ER mass and impaired ER-associated calcium homeostasis.

**Mitochondrial calcium and ATP citrate lyase (ACL) activation are reduced in BCR-KO Ramos cells**

The ER provides mitochondria with calcium, which is essential for the maintenance of mitochondrial function and energy balance (Bustos et al, 2017). Thus, we hypothesized that reduced calcium levels in the ER of the BCR-KO B cells could lead to diminished mitochondrial function. Indeed, we found mitochondrial calcium to

statistical significance. N = 5; *$P = 0.016$. **(H)** Flow cytometric analysis of Mito Tracker Red-CMXRos–stained WT and KO cells. Significance was determined using the Mann–Whitney test. N = 5. **(I, J)** WT and KO Ramos cells were cultured overnight with or without oligomycin. Cell death was determined using 7-AAD. **(I, J)** A representative experiment is shown in (I), summary of the performed experiments is shown in (J). Statistical significance was determined using the ANOVA test. N = 7; ****$P < 0.0001$. **(K)** WT and BCR-KO cells were treated overnight with oligomycin. Global metabolic profiles were determined using UPLC-MS/MS (Metabolon). Shown are cell count–normalized results for glucose and glucose-6-phosphate (G6P). Welch's two sample $t$ test was used to determine statistical significance. N = 4; *$P < 0.05$. Whisker plots show minimum and maximum values, the box extends from the 25th to the 75th percentile, the line shows the median. Metabolomic studies shown in Figs 3K, 7G, S2A and D, and S6C and D were performed in parallel and can be, thus compared directly. Circles in the shown graphs represent independent experiments. **(A, C)** For experiments in which technical replicates were measured in parallel (A, C) a circle represents the mean value of these replicates. **(A, B, C, D, E, F, G, H, I, J, K)** KO, BCR-KO Ramos cells (A, B, D, E, F, G, H, I, J, K), or Ig heavy chain–deficient DG75 cells (C), WT, corresponding wild type cells. gMFI, geometric mean fluorescence intensity.

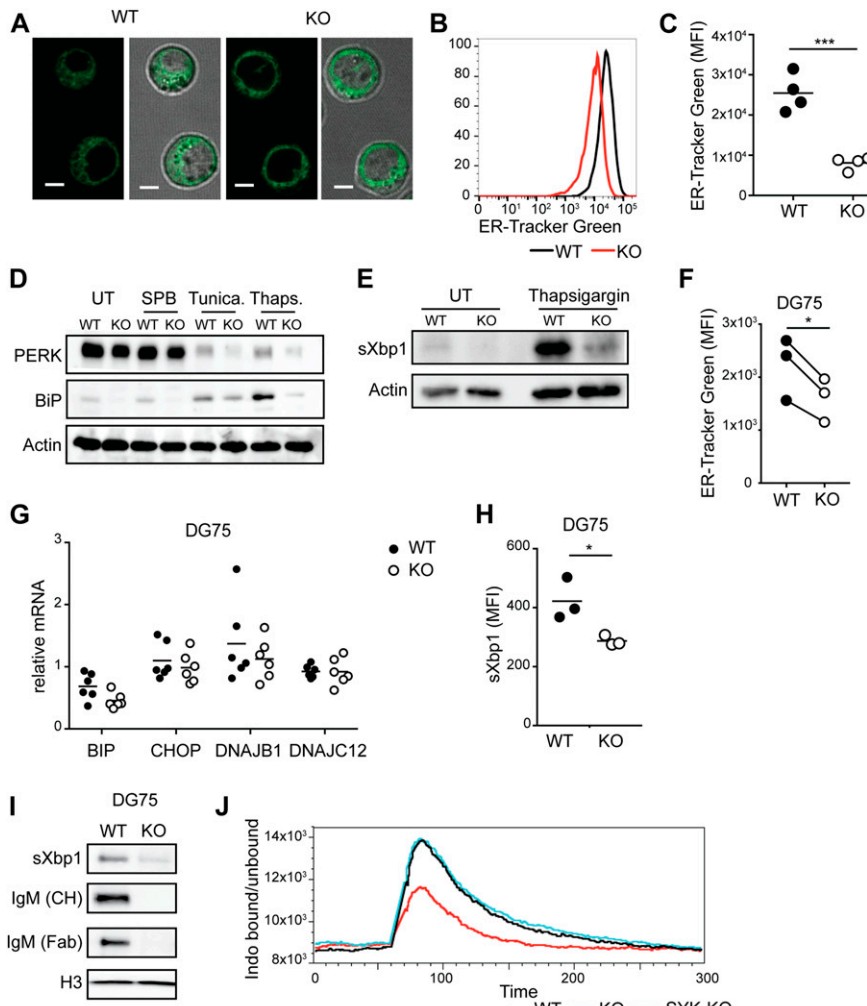

**Figure 5. ER expansion and function are impaired in BCR-KO B lymphoma cells.**
**(A)** Pictures show ER-Tracker Green stained cells alone (left panels) and overlaid with transmitted light (right panels) of WT and KO cells (magnification 63×, scale bar = 5 μm). **(B, C)** Ramos cells stained with ER-Tracker Green were analyzed by flow cytometry. **(B, C)** A representative experiment is shown in (B), and summary of four independent experiments is shown in (C). Significance was determined using unpaired *t* test. N = 4; ***P = 0.0004. **(D, E)** Cells were left untreated (UT) or were treated with the chemical chaperone 4-sodium phenyl butyrate (SPB) and the ER stress–inducing agents tunicamycin or thapsigargin overnight and assayed for the expression of the indicated proteins. Actin was used as a loading control. One of two independent experiments is shown. **(F)** DG75 cells were stained with ER-Tracker Green at day 6 after Ig heavy chain deletion. Significance was determined using the paired *t* test *P = 0.0262. **(G)** mRNA levels of the indicated genes in DG75 cells were analyzed on day 6 after Ig heavy chain deletion. **(H)** N = 6; (H) sXbp1 protein levels as determined by flow cytometry in single cell clones derived from DG75 cells. Significance was determined using the *t* test. N = 3; *P = 0.033. **(I)** Expression of the indicated proteins in DG75 single cell clones. Antibodies directed against the CH region and the Fab region of IgM were used. **(J)** Cells were loaded with Indo-1 AM and incubated with EGTA. Basal calcium levels were measured for 60 s. The cells were stimulated with thapsigargin. The ratio of bound/unbound Indo1 is shown. One of three experiments is shown. Circles represent independent experiments. **(G)** For experiments in which technical replicates were measured in parallel (G), a circle represents the mean value from these replicates. **(A, B, C, D, E, F, G, H, I, J)** KO, BCR-KO Ramos cells (A, B, C, D, E, J) or H–KO DG75 cells (F, G, H, I); WT, wild-type cells.

be reduced in BCR-KO Ramos cells in comparison with WT and Syk-deficient Ramos cells as indicated by staining with the fluorescent dye Rhod2-AM (Fig 6A). To assess the effect of reduced calcium levels on cell metabolism, we treated WT Ramos cells with the cell-permeable calcium chelator 1,2-Bis(2-aminophenoxy)ethane-N,N,N′,N′-tetraacetic acid tetrakis(acetoxymethyl ester) (BAPTA-AM) and measured oxygen consumption and medium acidification that are depend on metabolic functions in mitochondria and the cytosol, respectively. We found that only oxygen consumption (Fig 6B) but not the glycolytic flux (Fig S5A) was reduced after BAPTA-AM treatment, suggesting that intracellular calcium primarily affects mitochondrial function. Importantly, we confirmed that BAPTA-AM treatment does not induce cell death of Ramos cells (Fig S5B).

ER stress is known to lead to increased expression and activation of enzymes involved in fatty acid synthesis such as ACL supporting ER biogenesis (Basseri & Austin, 2012). Consistent with lower levels of ER stress, the BCR-KO Ramos cells have a lower ER mass (Fig 5B and C), a reduction of several components of fatty acid metabolism (Fig S2A), and impaired expression of phosphorylated and total ACL (Figs 6C and S5C). However, ACL activity is not only needed to support fatty acid synthesis but also can contribute to generation of

oxaloacetate, an important precursor for aspartate biosynthesis under conditions of ETC inhibition. To test whether BCR-KO cells are more sensitive to ETC inhibition, we treated the cells with the complex I inhibitor rotenone and found a stronger reduction in the viability of BCR-KO than WT Ramos cells (Fig 6D). Because pyruvate can be catabolized by pyruvate carboxylases to generate precursor molecules for aspartate, we provided the cells with an excess of pyruvate in an attempt to rescue cell death in BCR-KO cells after the inhibition of the respiratory chain. Indeed, pyruvate could partially rescue the survival of rotenone treated BCR-KO Ramos cells (Fig 6D). In summary, cellular processes governed by the ER such as mitochondrial calcium homeostasis or the expression of ACL are disrupted in BCR-KO lymphoma cells.

## Ig induced ER stress leads to increased oxygen consumption and metabolic flexibility

Our data suggest that decreased ER stress resulting from the lack of BCR components in the ER, rather than a lack of BCR/Syk-mediated signaling leads to metabolic impairments in BCR-KO Ramos cells. We hypothesized that reexpression of BCR components could rescue

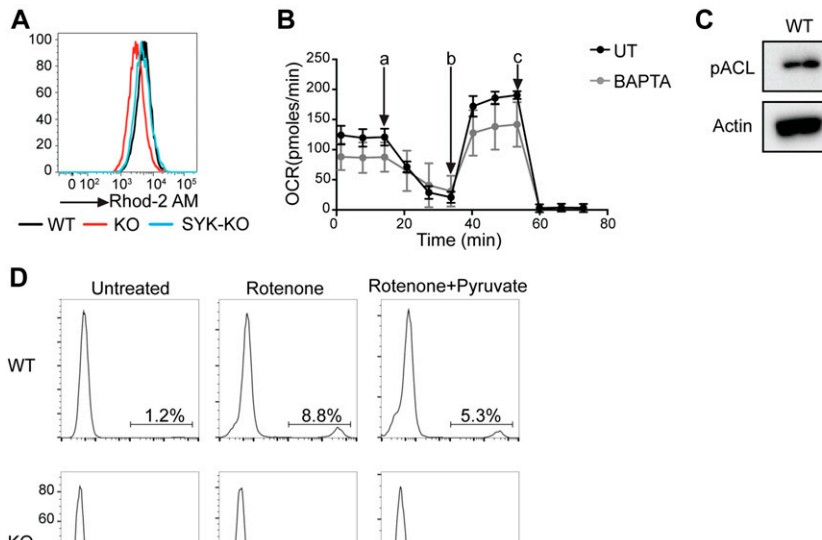

**Figure 6. Mitochondrial calcium and ACL activation are reduced in BCR-KO Ramos cells.**
**(A)** Cells were stained with Rhod-2-AM to assess mitochondrial calcium and analyzed by flow cytometry. One of six independent experiments is shown. **(B)** Cells were incubated with or without BAPTA-AM for 1 h and 30 min, and oxygen consumption was measured. The measurements were performed in technical triplicates and are displayed as mean ± SD. One of three independent experiments is shown. a, oligomycin; b, FCCP; c, rotenone + antimycin. **(C)** ACL phosphorylation was analyzed by Western blot. Actin was used as a loading control. One of four independent experiments is shown. **(D)** Cells were left untreated or were treated with rotenone or rotenone + pyruvate overnight. Cell death was determined by 7-AAD incorporation. One of six independent experiments is shown. KO, BCR-KO Ramos cells; WT, Ramos wild-type cells.

the phenotypes observed in BCR-KO Ramos cells. To verify our hypothesis, we reexpressed mIgM in BCR-KO Ramos cells (KO + mIg). These cells still lack the Igα/Igβ heterodimer and, thus, do not assemble a signaling competent BCR. Anti-IgM treatment or stimulation with cognate antigen did not induce calcium mobilization in these cells, confirming that they do not respond to BCR-induced activation (Fig S6A). Interestingly, in comparison with BCR-KO and WT, the KO + mIg Ramos cells had an increased cell mass and BiP production, indicating elevated ER stress (Fig 7A and B). Metabolomic profiling revealed that mIgM reexpression resulted in broad metabolic changes, including alterations in lipid and glutamine metabolism with several metabolites involved in fatty acid metabolism being strongly increased (Fig S6B and C). Moreover, basal oxygen consumption was significantly enhanced in the KO + mIg Ramos cells (Fig 7C and D) but glycolytic flux remained unchanged (Fig 7E). We previously showed that NAD+ levels are reduced in BCR-KO cells (Fig S2D). In contrast, NAD+ levels in KO + mIg Ramos cells exceeded those detected in WT cells (Fig 7F), which is consistent with their high levels of oxygen consumption. Furthermore, mIgM expression in the BCR-KO cells was sufficient to rescue survival after overnight oligomycin and rotenone treatment (Fig 7G).

To confirm that it is the Ig chain expression rather than ITAM signaling that is associated with ER stress and metabolic fitness, we also expressed the secreted form of IgM (sIgM) in BCR-KO cells (KO + sIgM) that does not associate with Igα and Igβ in normal cells and does not serve the function of a cell surface receptor. In comparison with the BCR-KO, the KO + sIgM Ramos cells displayed an increased BiP expression (Fig 7H), enhanced oxygen consumption (Fig 7I), and normal survival after oligomycin treatment (Fig 7J). The rescue of metabolic defects observed in BCR-KO Ramos cells by signal-incompetent mIgM and sIgM supports the notion that metabolic flexibility of Ramos WT cells depends on Ig expression rather than BCR signaling.

## Loss of Ig expression in naïve B cells results in a reduced ER mass

We next asked whether resting mouse B cells also require Ig production for normal ER homeostasis. Unlike lymphoma B cells, naïve B cells require the presence of the BCR and die upon H chain gene deletion unless pro-survival signals are provided (Srinivasan et al, 2009; Otipoby et al, 2015). Using B cells from mice, which contain a floxed Ig H chain variable region that can be deleted upon Cre enzyme expression, we confirmed that the BCR is needed for the maintenance of mature B cells (Fig 8A). As described before (Srinivasan et al, 2009), BCR-deficient cells (here referred to as IgM– cells) could be rescued by the expression of constitutively active PI3K (Fig 8A). At day 3, after the Cre-induced H chain gene deletion, the cell size of IgM– and IgM+ B cells was comparable (Fig 8B); however, the ER mass was significantly reduced (Fig 8C) in comparison with control B cells. Also, oxygen consumption was significantly reduced in primary B cells at day 6 after Ig H chain gene deletion as compared with controls (Fig 8D). Similar to lymphoma cells, resting IgM– B cells showed a reduced expression of mRNA encoding ER-associated proteins such as BiP or Edem (Fig 8E). To test whether pro-survival signals could rescue the expression of ER proteins in IgM– cells, we deleted the H chain gene in B cells expressing a constitutively active mutant of PI3K (P110*). Although BiP mRNA levels increased in P110* cells, BiP expression remained significantly lower in IgM– B cells in comparison with control B cells (Fig 8E). In a similar approach, we deleted the H chain gene in Bcl2 transgenic B cells and stimulated them with LPS. Upon LPS stimulation, B cells acquire cell mass to prepare for proliferation and differentiation to plasma blasts/plasma cells. This process is accompanied by an increase in the expression or ER-resident molecules such as BiP in normal cells (Fig 8F). In contrast, we found IgM– cells to be unable to blast (Fig 8G) and to fail to increase BiP expression, despite their ability to respond to LPS stimulation by increasing the expression of activation markers such as

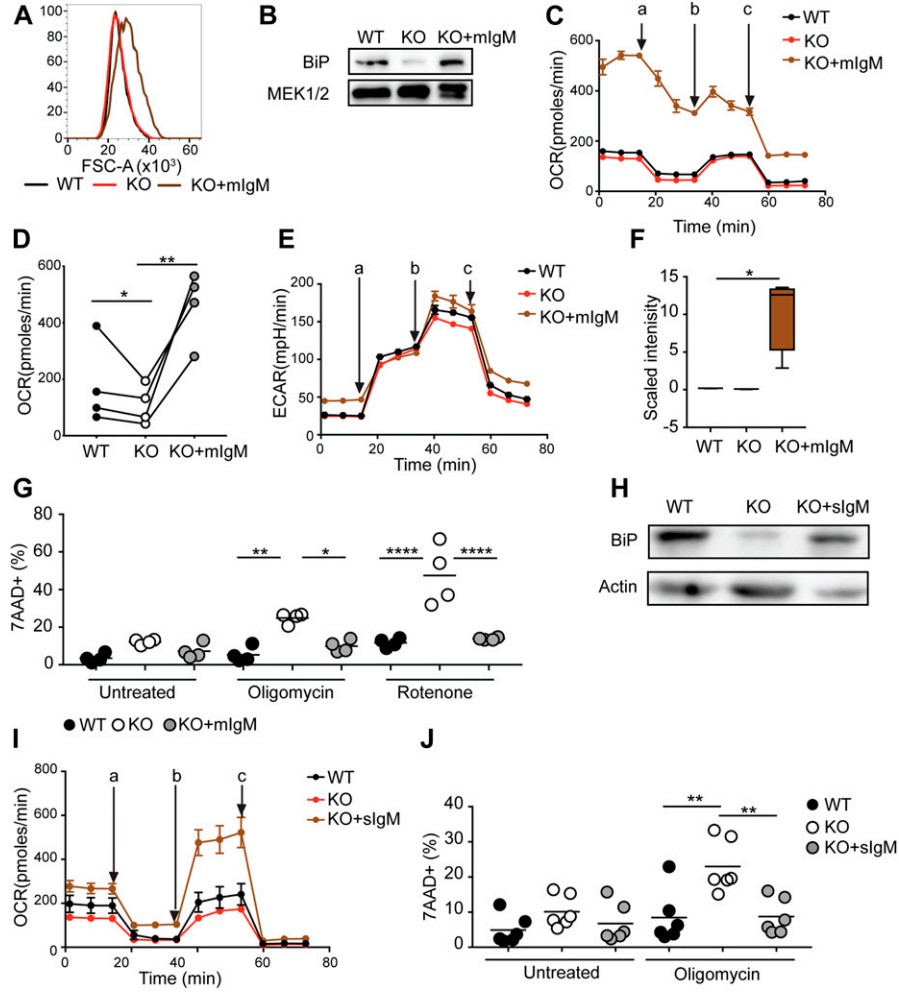

**Figure 7. Immunoglobulin induced ER stress leads to increased oxygen consumption and metabolic flexibility.**

**(A)** FSC-A as a measure of cell size was determined by flow cytometry. One of two experiments is shown. **(B)** Cells were assessed for the expression of the indicated proteins. MEK1/2 is used as loading control. One of three independent experiments is shown. **(C)** Cells were resuspended in medium containing pyruvate, glutamine, and glucose, and OCR was measured using Seahorse flux technology. The measurements were performed in technical triplicates and are displayed as mean ± SD. One of four independent experiments is shown. a, oligomycin; b, FCCP; c, rotenone + antimycin. **(C, D)** Summary of basal oxygen levels measured in experiments described in (C). Statistical significance was determined using the paired *t* test. N = 4; *P = 0.0442, **P = 0.0094. **(E)** Cells were resuspended in glucose-free medium containing glutamine, and ECAR was measured using Seahorse flux technology. The measurements were performed in technical triplicates and are displayed as mean ± SD. One of two independent experiments is shown. a, glucose; b, oligomycin; c, 2DG. **(F)** Global metabolic profiles of WT, KO, and KO + mIgM Ramos cells were determined using UPLC-MS/MS (Metabolon). Shown are cell count normalized results for NAD+. WT and KO results are the same samples as shown in S2D. Welch's two sample *t* test was used to determine statistical significance. N = 4; *P < 0.05. Whisker plots show minimum and maximum values, the box extends from the 25th to the 75th percentile, and the line shows the median. Metabolomic studies shown in Figs 3K, 7F, S2A and D, and S6B and C were performed in parallel and can be, thus, compared directly. **(G)** Cells were left untreated or were treated with oligomycin or rotenone overnight. Cell death was determined by 7-AAD incorporation. Significance was determined using the ANOVA test. For clarity, only comparisons of KO versus WT or KO + mIgM are shown. N = 4, *P = 0.0311, **P = 0.0021, ****P < 0.0001. **(H)** BiP expression was determined by Western blot. Shown is one of two experiments. **(I)** Cells were resuspended in medium containing pyruvate, glutamine, and glucose, and OCR was measured using Seahorse flux technology. One of two independent experiments is shown. a, oligomycin; b, FCCP; c, rotenone + antimycin. **(J)** Cells were cultured overnight with or without oligomycin. Cell death was determined using 7AAD. N = 6, **P = 0.0022 and 0.0028. Statistical significance was determined using the ANOVA test. For simplicity, only comparisons of WT and KO cells and of KO + sIgM and KO cells are shown. Circles represent independent experiments. **(C, E, I)** For experiments in which technical replicates were measured in parallel (C, E, I), a circle represents the mean value of these replicates. KO, BCR-KO Ramos cells; KO + mIgM, BCR-KO Ramos cells reconstituted with Ig heavy and light chain; KO + sIgM, KO cells reconstituted with the secreted version of IgM; WT, Ramos wild-type cells.

CD69, as reported before (Otipoby et al, 2015) and confirmed in our experiments (Fig 8H). Thus, despite Bcl2-induced pro-survival signaling in IgM− B cells, BiP expression remained dependent on the presence of Ig. In summary, these findings indicate that the BCR serves two functions in resting B cells: first, as a receptor mediating pro-survival signaling and second, as an inducer of proteins associated with normal ER homeostasis.

# Discussion

We here showed that the loss of the BCR resulted in reduced ER mass and function. Interestingly, the reduction of ER mass is associated with metabolic defects. Moreover, we demonstrate that ER homeostasis and metabolic function can be restored by expression of mIgM or sIgM in the absence of functional ITAM-bearing signaling

molecules, suggesting that the observed metabolic phenotype is not dependent on BCR signaling.

It has been previously shown that B-cell activation is coupled to ER expansion to accommodate the increased production of sIg during plasma cell differentiation (Gass et al, 2004). In this regard, excess production of proteins during plasma cell differentiation leads to ER stress (Ma et al, 2010; Bravo et al, 2013), which in turn activates signaling pathways triggering ER biogenesis to increase protein-folding capacity. Expression of ER stress–associated molecules such as sXbp1 is sufficient to drive ER expansion (Sriburi et al, 2007), and the expression of the Ig H chain is needed for Xbp1 splicing in stimulated B cells (Iwakoshi et al, 2003). Moreover, expression of secreted μ-chain is sufficient to induce ER expansion in Hela cells (Bakunts et al, 2017), suggesting that ER mass is dynamically adjusted in response to Ig production. Although ER stress responses are recognized as one of the driving forces of the plasma cell program (Gass et al, 2004; Lam & Bhattacharya, 2018), some of

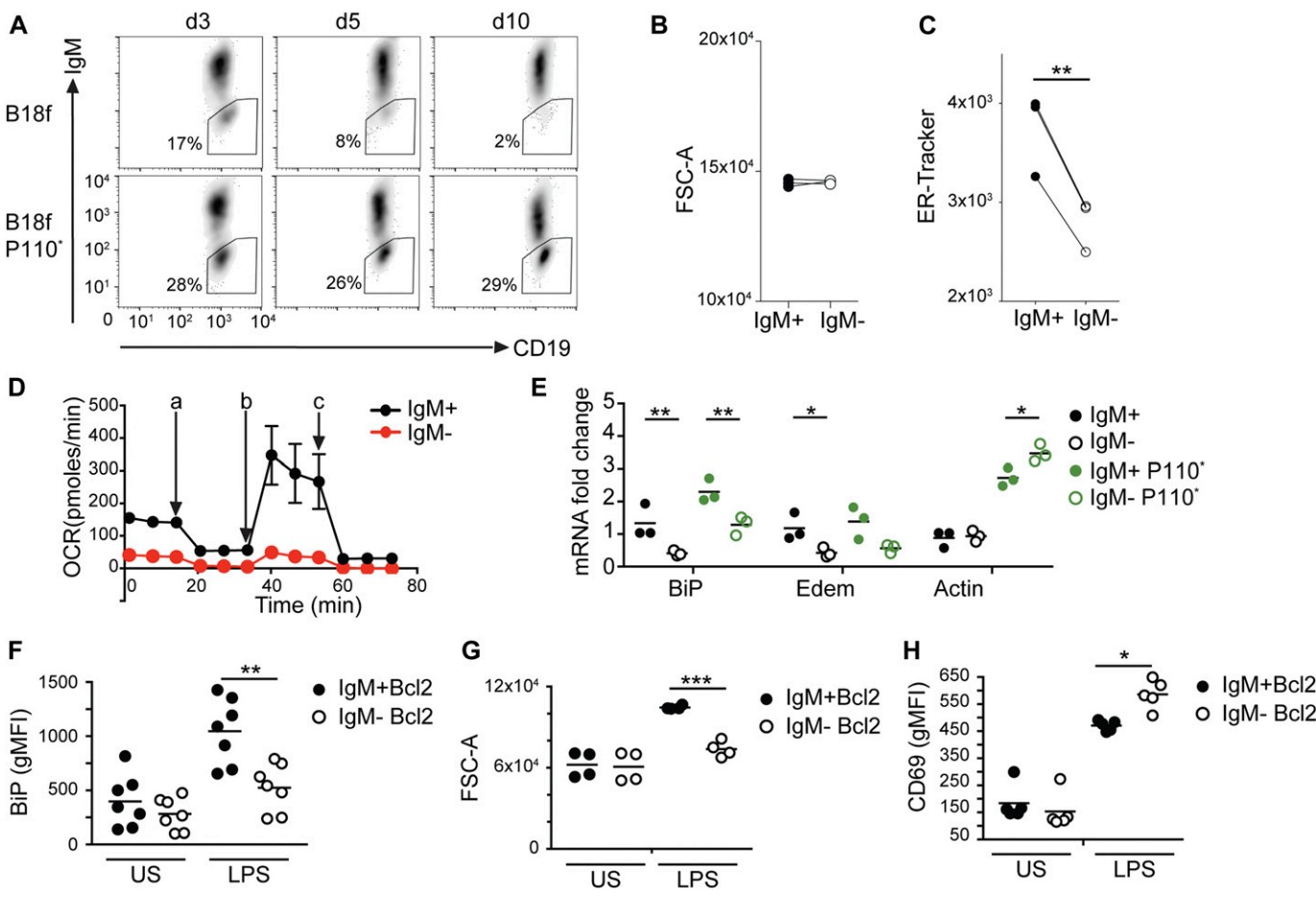

**Figure 8. Immunoglobulin expression is needed for ER maintenance in resting B cells.**
**(A)** Mature resting B cells from *B1-8^flox* (B18f) and *B1-8^flox* × *R26Stop^flox P110** (B18f P110*) mice were treated with TAT-Cre to induce the deletion of loxP-flanked sequences. The frequency of BCR-negative B cells was monitored over time. **(B)** Cell size of *B1-8^flox* × *Mb1-creERT2* B cells treated with tamoxifen to induce the deletion of the Ig heavy chain. **(B, C)** Cell size (B) and ER mass (C) of cells positive for surface IgM (IgM+) and negative for surface IgM (IgM−) was determined on day 3. Data were tested for significance using the paired *t* test. **(A, B)** N = 3; *P* = 0.75 (A), ***P* = 0.0087 (B). Experiments were performed in six technical replicates. Circles indicate mean values of these replicates obtained from three different mice. **(D)** Mature resting B cells from *B1-8^flox* × *EµBcl2* mice treated with TAT-Cre to induce the deletion of the Ig heavy chain, FACS-sorted on d5 upon TAT-Cre, and let to recover for 24 h post-sorting were resuspended in medium containing pyruvate, glutamine, and glucose, and OCR was measured using Seahorse flux technology on d6. The measurements were performed in technical triplicates and are displayed as mean ± SD. a, oligomycin; b, FCCP; c, rotenone + antimycin. **(E)** mRNA levels for the indicated genes were determined on day 2 after TAT-Cre–mediated deletion of loxP-flanked sequences in B18f and B18f P110* B cells positive or negative for surface IgM expression. Results were tested for significance using two-way ANOVA. For clarity, only differences between IgM+/IgM− and IgM+ P110*/IgM− P110* cells are shown. ***P* = 0.0078 and 0.0057, **P* = 0.0351 and 0.0402. Graph shows data obtained from three mice. **(F, G, H)** B cells from *B1-8^flox* × *EµBcl2* mice were purified, treated with TAT-Cre, kept for 4–6 d in culture to allow for IgM deletion, and stimulated overnight with LPS. **(F, G, H)** BiP protein levels (F), cell size (G), and CD69 surface levels (H) were assessed by flow cytometry. Results were tested for significance using ANOVA. For clarity only differences between IgM+ and IgM− cells are shown. Circles represent data obtained from different mice. If samples were measured in technical replicates, circles represent mean values obtained from these replicates. For F: N = 7, ***P* = 0.002; for G: N = 4, ****P* = 0.0006; for H: N = 5, **P* = 0.0192. Heavy chain deletion was performed in an in vitro cell culture in the presence of the cytokine BAFF.

the branches of the ER stress response can precede an increase in Ig production (Benhamron et al, 2015). Our findings suggest that mIg production in unstimulated B lymphoma or normal resting B cells are sufficient to drive ER expansion and the appropriate expression of ER-resident proteins such as BiP and sXbp1. In summary, ER mass is shaped by developmental cues and ER stress signaling in B cells and reflects the cells' need for protein folding capacity (Shaffer et al, 2004; Sriburi et al, 2007).

The ER plays a vital role in protein processing and quality control and has also been implicated to exert a multitude of different functions in regulating cellular metabolism. For example, the ER can affect mitochondrial function by governing cellular calcium homeostasis. Calcium is not only important for cytosolic signal

transduction but also plays various roles in regulating mitochondrial function. It has recently been shown that the inhibition of calcium transfer from the ER to mitochondria in cancer cells elicits diminished basal and maximal oxygen consumption, impaired ATP production, and increased activation of the mTORC1-inhibitor AMPK (Cardenas et al, 2016). Thus, perturbed calcium homeostasis may lead to impaired oxygen consumption and mTORC1 activity in both BCR-KO Ramos and IgM− naïve mouse B cells. Indeed, in addition to reduced ER-associated levels, we found calcium to be reduced in the mitochondria of BCR-KO Ramos cells and a reduction in calcium levels to be sufficient to inhibit oxygen consumption. In addition to calcium homeostasis, reduced ER biogenesis could impact cell metabolism in BCR-KO cells by an altered expression and activation

of enzymes involved in lipid synthesis. An essential prerequisite for ER biogenesis is lipid synthesis, and ER stress is known to induce expression of proteins involved in lipid metabolism. We found different components of fatty acid metabolism to be reduced and the expression of total and phosphorylated ACL to be decreased in BCR-KO Ramos cells. ACL is an extra-mitochondrial enzyme catalyzing the formation of acetyl-CoA, which is an essential component of fatty acid synthesis. LPS-induced differentiation of normal mouse B cells results in increased ACL enzymatic activity (Dufort et al, 2014). ACL inhibition leads to defective endomembrane expansion, proliferation, and plasma cell differentiation (Dufort et al, 2014). Thus, reduced ACL activity in BCR-KO Ramos cells could be a factor contributing to impaired ER expansion observed in these cells. Moreover, in addition to lipid generation, ACL can also support metabolic flexibility under conditions of impaired mitochondrial function. A key function of the mitochondrial ETC is to provide electron acceptors for aspartate synthesis, an amino acid needed for protein-, purine-, and pyrimidine-synthesis (Birsoy et al, 2015; Sullivan et al, 2015). Upon inhibition of complex I of the ETC with rotenone, cells can use alternative strategies to generate aspartate. For this, they need to have a source of oxaloacetate, which can be made from malate by malate dehydrogenases, from pyruvate by pyruvate carboxylase, or citrate by ACL (Birsoy et al, 2015). We found BCR-KO Ramos cells to be more sensitive to rotenone-induced inhibition of mitochondrial respiration than WT cells. Moreover, providing BCR-KO Ramos cells with increased levels of pyruvate partially rescued their survival. Thus, our data suggest that although BCR-KO Ramos cells are less efficient in supporting biosynthesis through ACL activity, this defect can be rescued by providing the cells with increased levels of pyruvate. In summary, the various metabolic defects observed in BCR-KO Ramos cells could be a consequence of disrupted ER homeostasis.

Previously, BCR-mediated signaling has been shown to support B lymphoma fitness and compounds targeting components of the BCR/Syk-dependent signaling pathway are currently used in the clinics (Niemann & Wiestner, 2013; Rickert, 2013; Young et al, 2015; Burger & Wiestner, 2018). Our study reveals that in addition to its central role as a receptor on the cell surface, the BCR can already affect B-cell metabolism at the stage of assembly in the ER. Importantly, this regulatory process is independent of the classical Syk-mediated signal. Supporting this notion, we show that Syk-deficient cells do not display disrupted calcium levels in the ER or the mitochondria and do not manifest metabolic defects. On the other hand, expression of signaling-incompetent immunoglobulins is sufficient to restore oxygen consumption in BCR-KO cells. Although our data show that the expression of the Ig H and L chain trapped in the ER can rescue some of the metabolic phenotypes in BCR-KO cells, it should be noted that an inability to resolve ER stress overall reduces cellular viability. B cells expressing a dysfunctional BCR, which is unable to reach the plasma membrane, are not desirable in the mature B-cell pool and need to be eliminated. Thus, while mild ER stress can increase metabolic fitness in B cells, these beneficent effects may be overshadowed by pro-apoptotic signals induced by high levels of ER stress. It may be the duration/magnitude of ER stress and the signaling context that ultimately tip the balance between adaptation or cell death.

In summary, we propose a model in which moderate levels of ER stress originating from nascent Ig chains support ER expansion, which in turn shapes B cell metabolism. Our study highlights the currently underappreciated role of the BCR in regulating cellular fitness through ER homeostasis. Targeting ER homeostasis might thus represent a novel treatment strategy for B cell lymphomas resistant to BCR-signaling inhibitors.

# Materials and Methods

## Cell lines and cell culture

WT and BCR-KO Ramos cells were generated as described (He et al, 2018). Syk-deficient Ramos cells were generated using CRISPR/Cas9. To reexpress the H and L chain, BCR-KO Ramos cells were retrovirally transduced with a construct encoding eGFP-rigid linker-4-hydroxy-3-iodo-5-nitrophenyl acetate (NIP)–specific mouse $\mu$-chain-P2A sequence-NIP–specific L chain. To generate Ramos cells expressing the secreted version of IgM, BCR-KO Ramos cells were retrovirally transduced with two plasmids: The first plasmid encoding the NIP-specific human $\mu$-chain with secretion sequence–P2A sequence–human lambda L chain–fusion of the yeast GCN4 leucine zipper and a C-terminal fragment of CFP, the second plasmid encoding a leucine zipper and an N-terminal fragment of YFP (Kohler et al, 2008). Combined, these two plasmids encoded for a fluorescent marker, which was used for selection. The cells were cultured in Roswell Park Memorial Institute (RPMI) medium (Thermo Fisher Scientific) containing 10% fetal bovine serum (Biochrom AG), 100 units/ml penicillin + 100 $\mu$g/ml streptomycin (Thermo Fisher Scientific) and 57 $\mu$M $\beta$-mercaptoethanol (Sigma-Aldrich). For cultures with different carbon donors, glucose-free RPMI (Thermo Fisher Scientific) supplemented with 10% dialyzed FBS (Sigma-Aldrich), 100 units/ml penicillin + 100 $\mu$g/ml streptomycin (Thermo Fisher Scientific), 57 $\mu$M $\beta$-mercaptoethanol (Sigma-Aldrich), and 10 mM glucose (Sigma-Aldrich), 10 mM galactose (Sigma-Aldrich), or 10 mM GlutaMAX, as indicated, was used.

DG75 lymphoma cell line stably expressing Cas9 was generated as described (Kabrani et al, 2018). CrispR-Gold software (Chu et al, 2016) was used to design three single guide RNAs (sgRNAs) targeting human IgM in CH1 region. sgRNA-targeting sequences were cloned into the BbsI sites of the MSCV_U6_CcdB_PGK_Puro_T2A_BFP vector as described before (Chu et al, 2016). The sgRNA showing highest efficiency of IgM deletion (sgRNA_hIgHM_CH1_f CACCGGA-TACGAGCAGCGTGGCCGT, hIgHM_CH1_r AAACACGGCCACGCTGCTCGTATCC) was selected for the study. 5 $\mu$g of pMSCV vector expressing sgRNA targeting human IgM in CH1 region was transfected into $5 \times 10^6$ DG75 cells in nucleofection buffer (5 mM KCl, 10 mM MgCl$_2$, 20 mM Hepes, 90 mM Na$_2$HPO$_4$/NaH$_2$PO$_4$, pH7.2, 24 mM sodium succinate) using Nucleofector System (Lonza) with program X-001. DG75 cells were cultured in RPMI (Gibco) containing 10% fetal bovine serum (Sigma-Aldrich).

## Mouse lines and isolation of primary resting B cells

B1-8$^{flox}$ (Lam et al, 1997) × E$\mu$Bcl2 mice (Strasser et al, 1996) B1-8$^{flox}$ × R26Stop$^{flox}$ P110* (Srinivasan et al, 2009) and B1-8$^{flox}$ × Mb1-creERT2 (Hug et al, 2014) compound mutant mice were used to obtain

## Primer sequences.

| Primer sequence | |
|---|---|
| Mouse | |
| H3_F | GTGAAGAAACCTCATCGTTACAGGCCTGGT |
| H3_R | CTGCAAAGCACCAATAGCTGCACTCTGGAA |
| Rplp0_F | TTCATTGTGGGAGCAGAC |
| Rplp0_R | CAGCAGTTTCTCCAGAGC |
| BiP (Hspa5)_F | TTCAGCCAATTATCAGCAAACTCT |
| BiP (Hspa5)_R | TTTTCTGATGTATCCTCTTCACCAGT |
| Edem_F | CTACCTGCGAAGAGGCCG |
| Edem_R | GTTCATGAGCTGCCCACTGA |
| β-actin_F | GGCTGTATTCCCCTCCATCG |
| β-actin_R | CCAGTTGGTAACAATGCCATGT |
| Human | |
| BiP_F | TCCTATGTCGCCTTCACT |
| BiP_R | ACAGACGGGTCATTCCAC |
| GAPDH_F | CCAGGTGGTCTCCTCTGACTT |
| GAPDH_R | GTTGCTGTAGCCAAATTCGTTGT |
| DNAJB1_F | GACCCTCATGCCATGTTTGC |
| DNAJB1_R | CCCATAGGGAAGCCAGAGAAT |
| DNAJC12_F | GAATGTCACCCAGACAAGC |
| DNAJC12_R | GAATGGCATCGACATCTG |

resting splenic B cells using autoMACS proSeparator (Miltenyi) with CD43 depletion beads. Primary B cells were plated at $2 \times 10^6$ cells/ml in DMEM (Gibco) containing 10% FBS (Sigma-Aldrich), 2 mM L-GlutaMAX, 2 mM sodium pyruvate, 2 mM Hepes (Gibco), 1× NEAA (Gibco), β-mercaptoethanol (Sigma-Aldrich), 10 μg/ml gentamicin (Lonza), and 100 ng/ml BAFF (Peprotech). Animals were kept in accordance with the German Animal Welfare Act and experiments were approved by the regional council.

### IgM depletion in primary B cells

Treatment with a Cre fusion protein containing a transactivator of transcription (TAT) sequence (TAT-Cre) was used to induce deletion of pre-arranged IgM (B1-8) in *B1-8^flox^ × EμBcl2* and for simultaneous activation of P110* expression and IgM deletion in *B1-8^flox^ × R26Stop^flox^ P110** B cells. Purified B cells were washed in PBS and transduced with 50 μg/ml TAT-Cre at a density of $5 \times 10^6$/ml for 45 min at 37°C in serum-free, animal-defined component-free medium (HyClone), prewarmed to 37°C. 1 μM 4-hydroxy tamoxifen treatment was used to induce the deletion of B1-8f in *Mb1-creERT2* B cells, where specified $10^6$ cells/ml primary B cells were stimulated with 20 μg/ml LPS for 24 h.

### Cell survival and cell growth analysis

To determine survival, cells were cultured overnight at $5 \times 10^5$ cells/ml. Inhibitors were used at the following concentrations: 1 μM oligomycin (Sigma-Aldrich or Agilent), 2.2 mM sodium pyruvate

(Thermo Fisher Scientific), 100 nM rapamycin (LC Laboratories), and 1 μM rotenone (Sigma-Aldrich). Cell death was determined by 7-AAD or LIVE/DEAD fixable yellow dead cell stain (Thermo Fisher Scientific) incorporation. Cell growth was assessed using the CCK-8 kit (Sigma-Aldrich). To this end, the cells were plated on d0 at a concentration of $2 \times 10^4$ cells/100 μl on three identical plates. On d1, d2, and d3, 10 μl of the CCK-8 solution were pipetted into each well of one plate, respectively. OD was measured at 450 nm after 2 h of incubation at 37°C using the Multiskan FC optical reader (Thermo Fisher Scientific). A sample containing no cells was used to measure the background OD. The value of the background sample was subtracted from all samples measured on that day.

### Flow cytometry and cell sorting

Single-cell suspensions were stained with fluorescently conjugated antibodies for 15 min in FACS buffer (PBS + 1% BSA + 0.09% NaN₃) washed twice and analyzed on BD Fortessa. Data were analyzed using FlowJo software. Primary mouse B cells were stained with anti-B220 (RA3-6B2), anti-CD19 (6D5), anti-IgM (II/41), and anti-CD69 (H1.2F3), and for cell sorting on Aria (BD), Fab fragment anti-IgM antibody (Jackson) was used. Human DG75 cells were stained with anti-IgM (MHM-88), anti-IgK (MHK-49), and anti-CD79b (Igbeta, CB3-1) purchased from BioLegend.

For intracellular flow cytometry, True-Nuclear Transcription Factor Buffer Set (BioLegend) was used according to the manufacturer's protocol, with exception of blocking and staining steps that were performed with perm/wash buffer supplemented with 2.5% BSA and 1% gelatin from cold fish (Sigma-Aldrich). Antibodies used were Zombie Acqua fixable viability dye 1:800 (BioLegend) anti-Xbp1s-PE 1:20 (Q3-695; BD Pharmingen), anti-BIP 1:200 (C50B12; Cell Signaling Technology), anti-rabbit APC 1:200 (Jackson), and PE mouse IgG1, K isotype control 1:40 (MOPC-21; BioLegend).

### Calcium flux measurement

An Indo-1 solution was prepared by mixing 50 μg Indo-1 AM (Biomol) with 115 μl FBS (Biochrom) + 25 μl DMSO (Sigma-Aldrich) and 25 μl Pluronic-F127 (AAT Bioquest). $0.6 \times 10^6$ or $1 \times 10^6$ cells were incubated in 1 ml RPMI (Thermo Fisher Scientific) + 1% FBS (Biochrom AG) + 15 μl Indo-1 solution at 37°C for 45 min. Cells were washed and used at a concentration of $10^5$ cells in 500 μl RPMI (Thermo Fisher Scientific) + 1% FBS (Biochrom AG) for the measurement. Basal calcium levels were acquired for 1 min. Subsequently, the cells were stimulated with 6 μl pervanadate solution (10 mM orthovanadate + 30% $H_2O_2$), thapsigargin (to a final concentration of 1 μM) or antimouse IgM (to a final concentration of 10 μg/ml), and NIP-BSA-biotin (to a final concentration of 20 ng/ml). If extracellular calcium needed to be removed, the cells were incubated in 0.5 mM EGTA (Fluka) containing medium 5 min before the measurement and for the duration of the experiment. Data were acquired on an LSR II flow cytometer (BD) at the Lighthouse Core Facility of the University Clinic Freiburg and the Max Planck Institute for Immunobiology and Epigenetics.

### Metabolic flux analysis

To assess glycolysis, the cells were resuspended in 50 μl Seahorse XF Base Medium (Agilent) supplemented with 2 mM L-glutamine

(Thermo Fisher Scientific) and incubated for 30 min at 37°C in a $CO_2$-free incubator. Subsequently, additional 130 $\mu$l medium were added, and the cells were incubated for 1 h. ECAR was measured using the Seahorse XFe96 metabolic flux analyzer (Agilent). The cells were sequentially treated with 10 mM glucose (Sigma-Aldrich), 1 $\mu$M oligomycin (Agilent), and 30 mM 2DG (Sigma-Aldrich). Glycolytic flux was calculated by subtracting basal ECAR levels from ECAR levels obtained after glucose addition. Glycolytic capacity was calculated by subtracting ECAR levels induced by glucose addition from ECAR levels measured after oligomycin treatment. To assess mitochondrial function two different tests were performed. In the first test, the cells were resuspended in 50 $\mu$l Seahorse XF Base Medium (Agilent) supplemented with 2 mM L-glutamine (Thermo Fisher Scientific), 1 mM sodium pyruvate (Thermo Fisher Scientific), and 10 mM glucose (Sigma-Aldrich) and incubated for 30 min at 37°C in a $CO_2$-free incubator. Subsequently, 130 $\mu$l medium were added and cells were incubated for an additional 1 h. For experiments in which intracellular calcium levels were lowered, 25 $\mu$l of 25 $\mu$M BAPTA-AM (Sigma-Aldrich) and 105 $\mu$l medium were added instead of pure medium. The cells were sequentially treated with 1 $\mu$M oligomycin (Agilent), 1 $\mu$M FCCP (Agilent), and 1 $\mu$M rotenone + antimycin (Agilent). OCR values obtained after FCCP treatment represent maximal oxygen consumption, and spare respiratory capacity was determined as the difference between OCR levels obtained after FCCP treatment and basal oxygen consumption. In the second test, the cells were resuspended in 50 $\mu$l MAS buffer (70 mM sucrose, 210 mM mannitol, 5 mM Hepes, 1 mM EGTA, and 0.5% fatty acid–free BSA, pH7.2) and incubated for 30 min at 37°C in a $CO_2$-free incubator. Subsequently, 130 $\mu$l MAS buffer + 20–25 ng/ml saponin were added and the cells were incubated for an additional 1 h. The cells were sequentially treated with 4 mM sodium pyruvate (Thermo Fisher Scientific) + 2 mM malic acid (Sigma-Aldrich), 2 mM ADP (Sigma-Aldrich), 1 $\mu$M oligomycin (Sigma-Aldrich), and 1 $\mu$M rotenone (Sigma-Aldrich). OCR was measured using the Seahorse XFe96 metabolic flux analyzer (Agilent).

## Analysis of metabolic parameters

To assess mitochondrial calcium, $10^5$ cells were stained in 100 $\mu$l of 0.1 $\mu$g Rhod2-AM (Thermo Fisher Scientific) + 0.2% Pluronic-F127 (AAT Bioquest) in Ramos medium for 30 min at 37°C. The cells were washed with medium, incubated for 30 min in 500 $\mu$l Ramos medium at 37°C, and washed again before measurement. To measure glucose uptake, $10^5$ cells were stained in 100 $\mu$l 50 $\mu$M 2NBDG (Cayman Chemical) in PBS (Invitrogen) for 30 min at 37°C, a separate set of samples was kept in PBS (Invitrogen) for 25 min and stained with 2NBDG for the last 5 min of incubation. Both sets of samples were washed before measurement. To assess total ROS levels, $10^5$ cells were stained in 100 $\mu$l 10 $\mu$M H2DCFDA (Thermo Fisher Scientific) in PBS (Invitrogen) for 20 min at 37°C and washed before measurement. To stain the ER cells, $10^5$ (RAMOS), 2–5 × $10^5$ (DG75), and $10^6$ (primary B) cells were stained in 100 $\mu$l 0.33 $\mu$M (RAMOS), 0.5 $\mu$M (DG75), and 0.25 $\mu$M (primary B) ER-Tracker Green in HBSS (0.137 M NaCl, 5.4 mM KCl, 0.25 mM $Na_2HPO_4$, 0.44 mM $KH_2PO_4$, 1.3 mM $CaCl_2$, 1.0 mM $MgSO_4$, and 4.2 mM $NaHCO_3$) for 15 min (Ramos) or 30 min (DG75, primary B) at 37°C and washed twice before measurement. To analyze protein abundance, $10^5$ cells were stained in 100 $\mu$l 5 $\mu$M

eFluro670 (eBioscience/Thermo Fisher Scientific) for 5 min at room temperature, washed with Ramos medium, incubated in 500 $\mu$l Ramos medium for 5 min at room temperature, and washed again before measurement. To stain for mitochondrial mass, $10^5$ cells were stained in 100 $\mu$l 60 nM Mito Tracker-RedCMXRos (Thermo Fisher Scientific) in Ramos medium for 30 min at 37°C. The cells were washed before measurement. The cells were analyzed using an LSR II (BD) or a CyAn (Beckman Coulter) flow cytometer or an LSM 780 Microscope (Zeiss) at the Max Planck Institute for Immunobiology and Epigenetics or the BIOSS Centre For Biological Signalling Studies in Freiburg, Germany.

## Metabolomic profiling

For metabolomic profiling, the cells were plated at a concentration of 5 × $10^5$ cells/ml with or without oligomycin, incubated overnight, washed with PBS, and snap-frozen in liquid nitrogen. The samples were processed and analyzed by Metabolon Inc. The extracts were divided into five fractions: two for analysis by two separate reverse-phase (RP)/UPLC-MS/MS methods with positive ion mode electrospray ionization (ESI), one for analysis by RP/UPLC-MS/MS with negative ion mode ESI, one for analysis by hydrophylic interaction chromatography (HILIC)/UPLC-MS/MS with negative ion–mode ESI, and one sample was reserved for backup. Raw data were extracted and compounds were peak identified by Metabolon Inc. Missing values were imputed with the observed minimum after normalization.

## Gel electrophoresis and Western blot

Ramos cells were lysed for 10 min on ice using RIPA buffer (150 mM NaCl, 1% NP-40, 0.5% sodium deoxycholate, 0.1% sodium dodecyl sulfate, 50 mM Tris, and 1 mM EDTA). Sodium orthovanadate (1 mM), sodium fluoride (10 mM), and a proteinase inhibitor cocktail containing AEBSF, aprotinin, bestatin hydrochloride, E-64, EDTA, and leupeptin (Sigma-Aldrich) were added to inhibit phosphatase and proteinase activity. Protein content was determined using the BCA kit (Thermo Fisher Scientific), and equal protein amounts were loaded on a 10% polyacrylamide gel. Proteins were transferred to a polyvinylidene difluoride (PVDF) membrane (Millipore) and blocked with 5% milk powder in TBS+ 0.1% Tween. Following primary antibodies were used: anti-human IgM (Southern Biotech), diluted to 1:1,000; antihuman lambda (Southern Biotech), 1:1,000; antihuman Ig$\beta$ (eBioscience/Thermo Fisher Scientific), 1:1,000; and anti-human Ig$\alpha$ (BioLegend) 1:1,000. All other primary antibodies were obtained from Cell Signaling Technology and used at a 1:1,000 dilution in TBS+ 0.1% Tween. The membrane was incubated overnight with primary antibodies, washed three times with TBS+ 0.1% Tween, and incubated with horseradish peroxidase–coupled species-specific secondary antibodies for 2 h at room temperature. Clarity Western ECL substrate (Bio-Rad) was used to detect horseradish peroxidase.

DG75 cells were lysed in RIPA buffer (50 mM Tris, pH 8.0, 150 mM NaCl, 1% Triton-X, 0.5% sodium deoxycholate, and 0.1% SDS) containing 1× protease/phosphatase inhibitor (CST) for 30 min at 4°C in rotation, and debris was removed by centrifugation. Proteins from 2 × $10^5$ cells were loaded to a precast 4–15% gradient gel (Bio-Rad) and transferred to PVDF membranes using Trans-Blot Turbo Transfer System (Bio-Rad). Primary antibodies anti-IgM 1:300

(MHM-88; BioLegend), anti-fab region of human IgM 1:3,000 (MA2; Novus), anti-Xbp1s 1:500 (Poly6195; BioLegend), and anti-H3 1:1,000 (Cell Signaling Technology) were incubated in 5% milk in TBS + 0.1% Tween and anti-BIP 1:1,000 (C50B12; Cell Signaling Technology) in 2.5% BSA+1% gelatin from cold fish (Sigma-Aldrich) overnight at 4°C. For signal detection, SuperSignal West Femto (Thermo Fisher Scientific) was used.

### NAD(P)H fluorescence lifetime imaging

For NAD(P)H fluorescence lifetime imaging experiments, Ramos cells were handled as described above and kept on a heating plate at 37°C during the whole experiment. The fluorescence lifetime of the cofactors NAD(P)H was measured in these samples using a two-photon microscope equipped with a time-correlated single-photon counting system (LaVision Biotec). To assess NAD(P)H fluorescence, lifetime NAD(P)H in the cells was excited at 760 nm by a fs-pulsed Ti.Sa laser and their fluorescence was time-resolved detected at 460 ± 60 nm, as previously described (Bird et al, 2005; Niesner et al, 2008). The fluorescence decay curve encompassed 10 ns with a time resolution of 55 ps and is a multi-exponential function containing the mono-exponential decays of free NAD(P)H and of NAD(P)H bound to various enzymes.

Data evaluation was performed using the phasor approach (Digman et al, 2008; Leben et al, 2018). The fluorescence decay in each pixel of the image is Fourier-transformed at a frequency of 80 MHz and normalized, resulting into a phasor vector, with the origin at (0;0) in the cartesian system and the end at the position represented in the phasor plot. If the vector ends directly on the half-circle with the center in (0.5;0.5) and with a radius of 0.5, the fluorescence decay is mono-exponential, that is, pure substance. On the half-circle, vectors ending at large abscise values and low ordinate values correspond to short lifetimes and those ending at low abscise and ordinate values to very long lifetimes. The distribution of the fluorescence lifetimes on the half-circle is not equidistant but naturally logarithmic.

The phasor vector of a two-exponential decay of a mixture of two pure substances ends on the segment connecting the respective fluorescence decays of the pure components on the half-circle. The nearer the mixture phasor point on one of the pure substance phasor points is located, the higher is the relative concentration of this to the mixture. Similarly, a mixture of three components results in a three-exponential decay, and the phasor vector ends in the middle of the triangle, connecting the three phasor points of the pure substances. This scheme holds true for as many components as the analyzed mixture contains.

The pixel-based phasor plots represent the frequency of the phasor vector ending points for each pixel in the image showing a fluorescence signal. For the cell-based phasor plots, the images were segmented using the "particle analysis" and corresponding thresholding in FIJI/ImageJ. For each object, that is, cell, the average of the abscise and ordinate value, respectively, were calculated and displayed in the cell-based phasor plot. In addition, the average NAD(P)H fluorescence lifetime was also determined for each cell.

### qRT-PCR

Total RNA from sorted cells was extracted using Micro (<0.5 × 10$^6$ primary B cells) or Mini RNeasy Kit (QIAGEN GmbH) with on-column DNAse digestion. The cDNA was synthetized using SuperScript IV (Thermo Fisher Scientific), and 10–50 ng cDNA was used for amplification with Fast SYBR Green Master Mix on StepOne Plus (Applied Biosystems). $2^{-\Delta\Delta CT}$ method was used to quantify mRNA expression.

### Statistical analysis

Statistical analysis was performed using GraphPad Prism. The Shapiro–Wilk normality test was used to test for normal distribution. If data passed the normality test, the t test was used, otherwise the Mann–Whitney U test was used. Paired t test was used for linked normally distributed data. For multiple comparisons, ANOVA with Tukey's correction for multiple comparisons was used. Linear regression analysis was performed for samples repeatedly measured for several days. Tests used for analysis are indicated in the respective figure legends. Statistical analysis of data obtained from metabolic profiling was performed by the Metabolon company. Number of independent repeats and technical replicates is indicated in the respective figure legend. Experiments were considered independent from each other if cells were harvested and processed on separate days. For experiments performed with the Ramos cell line, one clone described by He et al (2018) was used (He et al, 2018). For experiments performed with DG75 cells, bulk DG75 cells with acute IgH deletion were used and subsequently one IgH-KO and WT clone were generated and used. For experiments with mouse B cells, experiments were considered independent if cells were obtained from different mice.

## Data Availability

Source data of the described metabolome analysis are deposited in the MetaboLights repository under the study identifier MTBLS186. https://www.ebi.ac.uk/metabolights/MTBLS186.

## Supplementary Information

## Acknowledgements

We would like to thank Eleni Kabrani for providing us with the Cas9-expressing DG75 cell line and Cornelia Gottwick and Jianying Yang for their help with cloning the secreted IgM construct and for providing the needed plasmids. We would like to acknowledge the Signalling Factory of the Research Centres BIOSS Centre for Biological Signalling Studies and Centre for Integrative Biological Signalling Studies (CIBSS) for providing support with flow cytometry. J Jellusova was supported by the Ministry of Science, Research and the Arts Baden-Wuerttemberg and the European Social Fund through a Margarete von Wrangell fellowship. This study was supported by the German Research Foundation (DFG) under Germany's Excellence Strategy (BIOSS-EXC 294 and CIBSS-EXC-2189, Project ID 390939984) and the TRR130 (TP-25 to J Jellusova, TP-02 to M Reth, and TP-C01 to R Niesner and AE Hauser). In German, J Jellusova wird gefördert durch die Deutsche Forschungsgemeinschaft (DFG) im Rahmen der Exzellenzstrategie des Bundes

und der Länder–EXC-2189 – Projektnummer 390939984. This project has received funding from the European Union's Horizon 2020 research and innovation program under the Marie Sklodowska-Curie grant agreement No 752747 (to M Caganova).

## Author Contributions

H Jumaa: resources, investigation, and methodology.
M Caganova: resources, investigation, and methodology.
EJ McAllister: investigation.
L Hoenig: resources and investigation.
X He: resources.
D Saltukoglu: resources.
K Brenker: resources.
M Köhler: investigation and methodology.
R Leben: investigation and methodology.
AE Hauser: supervision, investigation, and methodology.
R Niesner: formal analysis, supervision, investigation, and methodology.
K Rajewsky: conceptualization, resources, and supervision.
M Reth: conceptualization, resources, and writing—review and editing.
J Jellusova: conceptualization, resources, formal analysis, supervision, funding acquisition, investigation, methodology, project administration, and writing—original draft, review, and editing.

## Conflict of Interest Statement

The authors declare that they have no conflict of interest.

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
