## [Reviewer comments · Life Science Alliance]

Life Science Alliance

Immunoglobulin expression in the endoplasmic reticulum shapes the metabolic fitness of B lymphocytes

Huda Jumaa, Marieta Caganova, Ellen McAllister, Laura Hoenig, Xiaocui He, Deniz Saltukoglu, Kathrin Brenker, Markus Köhler, Ruth Leben, Anja Hauser, Raluca Niesner, Klaus Rajewsky, Michael Reth, and Julia Jellusova

DOI: <https://doi.org/10.26508/lsa.202000700>

Corresponding author(s): *Julia Jellusova, Albert Ludwigs University Freiburg*

Review Timeline:	Submission Date:	2020-03-13
	Editorial Decision:	2020-04-01
	Revision Received:	2020-04-10
	Accepted:	2020-04-15

Scientific Editor: Andrea Leibfried

Transaction Report:

Please note that the manuscript was previously reviewed at another journal and the reports were taken into account in the decision-making process at Life Science Alliance.

Referee #1:

In this manuscript, the authors investigate how BCR expression regulates the metabolism of B cells. The study is based on the characterization of Ramos cells, Ramos cells deficient for BCR or overexpressing the BCR. The authors found that BCR deficient Ramos have decreased metabolic function probably due to complex I deregulation. The authors demonstrate that the phenotype is not a consequence of BCR signaling but may result from decreased immunoglobulin expression within the ER. They mention that component of the ER -stress response such are reduced in BCR deficient cells and show that response upon treatment with ER-stress inducing drugs such as thapsigargin are enhanced in presence of the BCR. They propose that immunoglobulins-production mediated ER-stress could contribute to the metabolic response in these cells. While this concept is of interest and the data are well presented, the significance of this observation and the contribution of the ER-stress pathways should be further explored.

We thank referee 1 that she/he finds our study of interest and the data well presented

1) The study focuses on Ramos cells. Metabolism and ER-stress responses are susceptible to genetic deregulations and environmental factors. It would be important to verify these observations in another model. It would be particularly important to address the contribution and relevance of this pathway in an in vivo model.

We agree with referee 1 and have confirmed our main findings (Expression of ER associated proteins, ER expansion, mitochondrial respiration) also in a second human Burkitt lymphoma cell line (DG75) as well as in primary murine B cells. Thus, the discovered connection between Ig-production, ER homeostasis and metabolic fitness seem to be a conserved and important feature of B cell biology.

2) The authors should investigate the IRE1 XBP1 pathway in more details. This pathway is well known to contribute to B cell differentiation independently of its involvement in the ER-stress response. The author should delete XBP1 and IRE1 to address the contribution of RIDD and/or XBP1 to the phenotype. They could compare the results to PERK-deficient cells.

Our study is focused on the metabolic dysregulation in BCR-KO B cells and not per se on the ER stress response pathway in these cells. To study the IRE1 XBP1 pathway and the contribution of RIDD and PERK in these cells would require a whole set of new experiments which would delay and change the focus of the manuscript. However, in future experiments we will follow up the suggestions of referee 1 provided that sXbp1, BiP or PERK deficient Ramos B cells are viable.

3) In line with the above comment it would be important to add figure EV 4 to the Figure 6 and complete Figure 6 D with a panel investigating IRE1 and PERK activation (phostaq would be suitable here), XBP1 splicing and ATF4 translation. In Figure 6D the differences in expression between KO and WT of PERK or BIP are modest/not convincing in contrast to the interpretation made in the text. The authors should use a more quantitative method to assess activation of the pathways such as monitoring of ERDJ4 and GADD34 by qRT-PCR.

We have now combined EV4 and Figure 6 (see E in new Figure 5). We also have added data from a new cell line (DG75) and primary murine B cells and used flow cytometry and qRT-PCR to measure the expression of different ER associated proteins (see G,H,I in new Figure 5 and E and F in new Figure 8).

4) How these ER pathways contribute to the metabolic deregulation is not addressed in the

manuscript.

As explained above our study is focused on the metabolic dysregulation in BCR-KO B cells and not on the ER stress response

Minor points: 5) The y axes labels are missing in Fig 3E and Fig 3F

We have added the labels

6) The activation of ER-stress response pathways does not always reflect the presence of ER-stress. As mentioned above some these pathways have a function as differentiation factors including in B cells. I would recommend to rephrase some of these notions or better investigate the defects in ER homeostasis

We agree with referee 1 that the word “ER stress” might be misleading in describing the phenotype of BCR-KO Ramos cells and thus use the term disrupted or dysregulated “ER homeostasis” in the new version of our manuscript. We are using the word “ER stress” only when referring to Igs trapped in the ER or thapsigargin induced ER stress.

Referee #2: Review Jumaa et al.

The manuscript by Jumaa et al. aims to address an interesting question that pertains to fundamental cell biology and immunology: How is the integrity of the B cell as a sentinel of the immune system preserved? B cells should display an intact and functional B cell receptor (BCR), but should perish if they do not display the BCR. How do B cells that do not display the BCR get eliminated? To shed new light on this question Jumaa et al. exploited a B cell line (Ramos) that lacks the four components of the BCR (Ig-alpha, Ig-beta, the membrane-bound heavy chain (Ig-Hm), and the light chain (IgL)) - further referred to as 4KO. In the paper, the authors compare various characteristics of WT vs 4KO Ramos cells.

We thank referee 2 that she/he finds that our study addresses an interesting open question that pertains to fundamental cell biology and immunology.

Strangely, the authors seem to be surprised in fig 1 to find that the 4KO cells proliferate about as well as WT Ramos. Obviously, if a clone has been picked up upon ablation of all the BCR components, then this clone is viable. Surely, the 4KO has undergone some adaptation in order to sustain viability. For the remainder of the manuscript, the authors therefore investigate not only how cells behave in presence or absence of the BCR, but also how the adaptations that the 4KO underwent upon ablation may have led to differences in metabolism etc., etc.

Yes, we were surprised that the 4KO Ramos cells proliferate about as well as WT Ramos as it was assumed that B cell Lymphomas require continuous BCR expression and signaling for their tumor growth. We agree with referee 1 that developing normal B cells as well as B cancer cells can adapt to the loss of a gene but this ability to adapt has clearly its limitations. We have generated several other gene KO that do not allow the expansion of viable Ramos cells. Adaptation or not, important in this context is that we not only conducted a loss-of-function but also a gain of function experiment and showed that the re-expression of the heavy and light chain rescues the metabolic phenotype of BCR-KO Ramos cells. To support our statement that the expression of the heavy and light chain is sufficient to dramatically alter the metabolic profile of Ramos cells, independently of their function as signaling molecules, we have now performed a comprehensive analysis of the metabolome of these cells in comparison to the WT and BCR-KO

cells and added the results to the manuscript.

Furthermore, we have confirmed our main findings (Expression of ER associated proteins, ER expansion, mitochondrial respiration) also in a second human Burkitt lymphoma cell line (DG75) as well as in primary murine B cells. Thus, the discovered connection between Ig-production and ER stress seem to be a conserved and important feature of B cell biology.

For this reason alone, I have strong doubts that any of the results in this paper are necessarily reporting on how the BCR provides a survival signal, and, thus in my opinion, this manuscript is unfit to be published. The paper provides moreover a patchwork of conjectures, and the overall conclusions are naïve to say the least. To point out but at a few of these:

Production of secretory Igs in B cells does not lead to endoplasmic reticulum (ER) stress, since the BCR signaling prepares the B cells for their task as plasma cells. Before stimulation with antigen, expression of BCR components in B lymphocytes is constitutive and therefore not stressful by essence. Thus, the claim that ER stress-driven oxygen consumption is necessary for mitochondrial prowess, and hence, for metabolic flexibility, which would explain how the BCR would provide a survival signal, verges on the absurd for too many reasons to list here.

We do not understand the arguments of the referee 2 in this context. While it is true that plasma cells re-shape their transcription profile and prepare for increased protein secretion before antibody production is started, this does not exclude the possibility that the folding of antibodies in the ER creates signals that are interpreted as “ER stress” and contribute to the maintenance of ER mass. Our data show that in the absence of the BCR or Ig the expression of proteins associated with the ER stress response pathway such as BiP, sXbp1 and PERK are reduced. We have confirmed these findings a second human Burkitt lymphoma cell line (DG75) as well as in primary murine B cells

Our findings are also in agreement with published observations stating that:

- increased production of proteins during plasma cell differentiation leads to the activation of specific ER stress response pathways (Ma et al., 2010)
- the expression of Ig heavy chain is needed for Xbp1 splicing in stimulated B cells (Iwakoshi et al. 2003)
- heavy chain expression is sufficient to induce ER expansion in HeLa cells (Bakunts et al., 2017)

We have included the discussion of these papers in the manuscript. We also do not understand the argument of the referee 2 why the constitutive expression of BCR components should prevent them to be regulated in the ER and connected to the ER stress response pathway? The work of an editor is constitutive but still can be stressful from time to time. However, as mentioned in our response to referee 1 our study is not focused on the ER stress response pathway but rather on the metabolic defect and we now use the term disrupted or dysregulated “ER homeostasis” when referring to the phenotype of BCR-KO Ramos cells. In the new version of our manuscript we are using the word “ER stress” only when referring to Igs trapped in the ER or thapsigargin induced ER stress.

In summary our manuscript shows for the first time that the BCR can affect B cell metabolism through other mechanisms apart from its function as signaling receptor on the cell surface. The

discovery of a BCR-ER homeostasis connection is important to consider whenever the metabolic phenotype of B cells with altered Ig expression is analyzed.

The authors should create cell lines in which components of the BCR can be deleted in a conditional manner and/or similarly restored (to assess whether any of the effects upon BCR ablation can be salvaged upon reconstitution - Importantly, expression levels in any reconstitution experiments should be near- endogenous moreover -). Otherwise the authors merely will describe that cells have adapted (potentially in various ways even, as the authors might find out if they'd study other 4KO clones) in order to survive (as this is the selective pressure that brought the constitutive KO cells about). These cell lines are bound to have stronger or less strong glycolytic expenditure, a more or less extended ER, higher or lower basal Ca²⁺ levels in the cytosol etc., etc., depending on the adaptation programs that have been invoked. Again, none of which necessarily reports on the original biological question.

As suggested by referee 2 we have now deleted the gene for a BCR component, namely the membrane-bound Ig heavy chain in an inducible manner in primary murine B cells. The study of these cells supports our conclusions obtained by the analysis of human B cell lines.

Referee #3:

This paper is a novel exploration of the mechanisms by which the BCR regulates cell survival and growth, particularly in the context of transformed B cells. This is an interesting and important topic - the BCR is essential for maintenance of normal B cells and is also providing a selective advantage to lymphoma cells. But particularly the laaer occurs through a number of mechanisms that do not always correspond to the ones operating in normal B cells. The authors focus here on antigen-independent, tonic activity of the BCR as observed in Burkia and GC-type diffuse large B cell lymphomas. It has been shown previously, that the surface BCR in these tumors induces CD19 dependent signaling through the PI3K-Akt axis to mTOR and GSK3, promoting metabolic fitness. The authors suggest that while the surface BCR mediates tonic signaling to mTOR, this is not sufficient to explain effects on metabolic fitness of these cells. This is based on experiments showing that a BCR null clone (lacking heavy and light Ig chains, CD79A and CD79B) grows relatively normally and so does a clone lacking SYK. The BCR null clone, but not the SYK deficient clone, has reduced mass of the ER and reduced capacity to release calcium from the ER, along with mitochondrial defects in oxidative phosphorylation and enhanced sensitivity to drugs inhibiting mitochondrial respiration. This could be related to the BCR normally inducing ER expansion and plasma cell differentiation, the full extent of which is blocked in the lymphomas. In contrast, the authors provide an intriguing suggestion that the presence of the Ig in the ER is required to maintain ER and mitochondrial homeostasis. Indeed, transfecting Ig heavy and light chains into the BCR null cells enhances expression of ER proteins, increases mitochondrial respiration and protects the cells from drugs inhibiting mitochondrial function.

This is an entirely new view on the mechanisms, by which the BCR can contribute to regulation of malignant B cell growth. Previous research focused entirely on the tyrosine kinase-based mechanisms of signaling from the surface. The results are thus interesting and could have implications for strategies to limit malignant B cell growth.

We thank referee 3 that she/he finds our study interesting and important for the better understanding and treatment of human malignant B cells.

Some issues, however remain unresolved, and I think the conclusions will need to be substantiated by additional experiments. At the moment, it is especially not clear, how the role of

the Ig in the ER relates to the function of the normal BCR, which will be assembling with the CD79 components and transported to the cell surface. As a possible alternative explanation, it seems plausible, that the BCR null cells adapted by reducing their ER so that their growth can be supported by the reduced mTOR signalling. Inducing ER stress by expression of the Ig leads to ER expansion and increased calcium and mitochondrial respiration, but the cells cannot cope without a corresponding increase in mTOR signals and they start dying as the authors show. If so, cell surface signaling is still an essential function of the BCR in this lymphoma. There are also a few technical limitations that should be addressed.

Main issues:

1. All the conclusions are based on comparisons of independently derived clones, one clone per genotype. Since the mutations affect metabolic fitness, it is possible that some of the phenotypes observed are clone-specific adaptations to their expansion and culture. For example, S6 phosphorylation does not seem to correlate with either the signaling or ER-based pathways in the different clones. For the transfectants with heavy and light chain, it is not specified how were the positive cells selected. An acute deletion or expression of the genes studied will help to establish a direct effect on the pathways in examination.

We have now studied the BCR-ER homeostasis connection in a second human Burkitt lymphoma cell line (DG75) as well as in primary murine B cells with an inducible BCR deficiency.

2. Previous studies showed that low metabolic fitness of BCR-deficient Myc-transformed cells manifests as reduced growth in competitive cultures, which is consistent with competition for nutrients. Clearly, knockouts of some signaling components, such as CD19 or CD79B, impair growth in competitive cultures. The authors do not use this assay in this manuscript and suggest that the BCR null and SYK deficient cells grow normally. It is thus difficult to understand if the effects the authors study are the same or separate from the previously studied growth defects mediated by CD79B and CD19. Comparison to CD79B or CD19 KO would be very helpful.

The cells we are using in this current study are identical to cells used in the study of He et al. 2018 showing that CD79B supports the proliferative fitness of Ramos cells. Differences in cell growth are primarily seen when competition experiments between BCR-KO and WT Ramos cells are performed. In our experiments, BCR-KO cells displayed a slight trend towards slower proliferation, however the difference was not statistically significant (Fig.1C). We have repeated our experiments using a second cell line (DG75) and now show growth competition experiments for this cell line in figure 1.

While CD79B expression alone can rescue the competitive disadvantage of BCR-KO Ramos cells in terms of proliferation, our preliminary results suggest that it cannot rescue the metabolic phenotype described in the current manuscript. Below are two experiments. The first one shows that BCR-KO cells re-expressing CD79B cannot survive after oligomycin treatment. The second experiment shows that oxygen consumption is also not rescued (however this experiment has been performed only once so far). We would prefer not to include these results in our manuscript since they might lead the reader to two possibly wrong assumptions:

- 1) CD79B signaling does not change the metabolic profile of lymphoma cells
- 2) The folding of CD79B in the ER is not sufficient to rescue metabolic defects in BCR-KO Ramos cells

It is likely that classical BCR signaling also affects B cell metabolism. The BCR signaling

pathway includes many regulators of cell metabolism such as Akt, GSK3, mTORC1 and others. The CD79B homodimer in connection to the CD19 coreceptor seem to be able to activate the PI-3 kinase-AKT pathway and thus could affect cell metabolism. However, as shown in the He et al. 2018 paper the CD79B homodimer is expressed **only in rather low amounts** on the CD79B transfected BCR-KO cells and its expression **seem to be sufficient for a providing a proliferation signal** but maybe too low to induce ER expansion and to rescue the metabolic phenotype of BCR-KO cells.

In summary we think that the BCR shapes cell metabolism in B cell lymphoma through two mechanisms -classical ITAM-dependent signaling and by altering ER homeostasis. The goal of our manuscript is to describe this second, to this date unexplored pathway. The phenotype of the Syk-deficient Ramos cells suggest that it is the expression rather than the signaling of the BCR that induces metabolic changes in Ramos cells. How CD79B-CD19 alters the proliferation but not the metabolism of Ramos B cells is an interesting open question beyond the scope of this manuscript.

Exp.1.: Cells were plated on day 0 with or without oligomycin and cell growth was measured on day 1 and day 2. Results were normalized to day1 untreated cells. Shown are results for day 1 and day2 (WT= wildtype, BCR-KO= BCR-KO cells transfected with an empty virus, BCR-KO+IgBeta= BCR-KO cells transfected with IgBeta)

Exp.2.: Oxygen consumption of WT Ramos cells, BCR-KO Ramos cells and BCR-KO Ramos cells reconstituted with IgBeta

3. The rescue experiments with the heavy and light Ig chain need to provide more information. The issue is that the Ig expression does not restore the WT phenotype, but rather changes various metabolic and signaling aspects of the null cells, complicating interpretation. For example, while the resistance to oligomycin and rotenone is restored, the respiration rates, calcium levels, cell size and S6 phosphorylation are all different from either the null and the wild type cells. Most importantly, it is not clear if the heavy and light chains restore normal cell growth. Fig EV5C suggests that they do not, as there is actually increased cell death during culture. This could mean that the restoration of the ER phenotype is not sufficient to rescue full growth, and that signaling competent BCR is still needed. Again, competitive cultures acutely after transfection could resolve these issues. The effect of the rescue on the calcium is particularly unclear, as figure 8B does not include comparison of the transfected cells to the null and WT cells. It is thus impossible to say if calcium homeostasis has any role in the rescue. Similarly, is ER mass rescued? Also, do the authors think the rescue is specific to the Ig, or would any ER stress also increase resistance to mitochondria inhibition?

We have included a metabolomics experiment to further the analysis of the metabolic profile of the BCR-KO cells reconstituted with the heavy and light chain. Indeed the expression of the heavy and light chain does not revert the phenotype to what is observed in the WT. This was expected considering that the expression of Ig heavy and light chain without CD79a and CD79b causes chronic ER stress and these cells are thus different than WT cells.

The main goal of the experiment was to show that Ig expression can induce broad metabolic changes independently of their signaling function. In many instances the reconstituted cells show the opposite phenotype to the BCR-KO cells. While oxygen consumption is decreased in BCR-KO cells in comparison to the WT, oxygen consumption is increased in the reconstituted cells in comparison to the WT. Similarly NAD⁺ levels and different lipid metabolites are decreased in BCR-KO cells in comparison to WT, but are increased in the reconstituted cells in comparison to WT. This suggests that BCR-KO cells that experience less “ER stress” than the WT cells show reduced mitochondrial function in comparison to the WT and the reconstituted cells that experience more “ER stress” than the WT show increased mitochondrial function. Calcium homeostasis could be one factor contributing to this phenotype. We have added an experiment where we treated WT Ramos cells with the calcium chelator BAPTA-AM to demonstrate that reducing intracellular calcium levels will lead to reduced mitochondrial function. However other ER dependent mechanisms may play a role as well, considering that ER expansion induces lipid synthesis and other metabolic process.

We have changed the discussion of our manuscript to clarify these issues.

4. It is not clear, how the effects of the Ig transfection into the BCR null cells relate to function of the normal BCR. Previously, a single KO of CD79B resulted in impaired growth (He et al, 2018), yet these cells have Ig in the ER. Is in this case the impairment occurring through a different mechanisms than the ER defect in the BCR null cells? What exactly is required in the ER to promote the calcium homeostasis? The authors should provide some explanation.

See above.

Minor issues: There seem to be very little difference in OCR between WT and KO cells in Fig8D, in contrast to Fig 3A. Legends for EV do not correspond to the figures. EV1 missing?

The difference between WT and KO in OCR in Fig8D (now 7C) is slightly dwarfed due to a need to use a scale that accommodates the reconstituted cells. However even if adjusting the scale, the difference is indeed not as dramatic as in the example shown in Fig3A. There seem to be day to day variations in the OCR of both WT and KO cells. Figure 3B provides a summary of 9 experiments and shows that the differences in oxygen consumption between WT and KO can vary. Statistical analysis of these results shows however that reduced oxygen consumption observed in KO B cells is statistically significant.

Supplementary figures have been changed.

April 1, 2020

RE: Life Science Alliance Manuscript #LSA-2020-00700-T

Dr. Julia Jellusova
Albert Ludwigs University Freiburg
Schänzlestraße 18
Baden Württemberg 79104
Germany

Dear Dr. Jellusova,

Thank you for submitting your revised manuscript entitled "Immunoglobulin expression in the endoplasmic reticulum shapes the metabolic fitness of B lymphocytes". Two of the original reviewers re-evaluated your work and appreciate the changes introduced in revision. We would thus be happy to publish your paper in Life Science Alliance pending final minor revisions:

- Please address the remaining minor concerns of rev#1
- Please include the supplementary methods in the main methods section to improve discoverability by the readers
- Please upload all figures, including supplementary figures, as individual files; the legends should all get moved into the main manuscript file
- Please add scale bars to Fig 5A and S3A
- Please use "p" when describing p-values in your figure legends
- Please move the Data and materials availability section to the methods section and rename it to "Data availability". The data should get deposited now and the accession code should get included in this section.
- Please make sure that all authors are listed in our submission system with functional email addresses and that they consent to the publication of the paper
- You will be prompted to fill in some mandatory fields in our submission system, please do

A. FINAL FILES:

B. MANUSCRIPT ORGANIZATION AND FORMATTING:

Sincerely,

Andrea Leibfried, PhD
Executive Editor
Life Science Alliance
Meyerhofstr. 1

69117 Heidelberg, Germany
t +49 6221 8891 502
e a.leibfried@life-science-alliance.org
www.life-science-alliance.org

Reviewer #1 (Comments to the Authors (Required)):

The authors show that deletion of the BCR impairs B cells not just by reducing of survival signalling from the plasma membrane, but also by limiting metabolic activity through effects on the ER, mitochondrial calcium and metabolic flexibility. Although this ER connection is not associated with cell death or impaired growth, I think it is worthwhile addition to the knowledge of B cell physiology and lymphoma pathology.

This manuscript is much improved from the previous version and is now much more convincing in showing the connection of the effects of the immunoglobulin in the ER to metabolism. Although the original work on the Ramos cells was limited by the independent derivation and adaptation of the BCR and SYK KO clones, the new data reproducing the phenotype in DG75 cells and in primary B cells show that the conclusions are valid across cellular models. Most importantly, the experiments with primary cells show that the deletion of the BCR heavy chain reduce ER size independently of rescued survival (by constitutively active PI3K or over-expression of BCL2). Thus expression of the BCR does have effects on the ER that are independent of canonical ITAM signalling.

Two minor points:

In Fig 8 it would be worthwhile to explain in text or legend that the B1-8 deletion is done in in vitro cultures with BAFF.

I can't find evidence that Ig is required for ER function or XBP splicing in the three references given on line 271: Bakunts, Orsi et al., 2017, Ma, Shimizu et al., 2010, Shaffer, Shapiro-Shelef et al., 2004. I think it is the Iwashima 2003 paper that shows that.

Reviewer #3 (Comments to the Authors (Required)):

The revised version of the manuscript is improved. In particular the fact that they could confirm the main findings in an other cell line is particularly important.

A few points were not addressed and considered out of scope: for exemple the study in not confirmed in an in vivo model and the investigation of other ER-signaling pathways such as RIDD to the ISR were not considered. These points would have straighten the message of the paper. However I feel that the study carry a significant message that is supported by the data presented.

We would like to thank the reviewers for re-evaluating our manuscript. Below are our responses to the Reviewer 1 comments.

Reviewer #1 (Comments to the Authors (Required)):

The authors show that deletion of the BCR impairs B cells not just by reducing of survival signalling from the plasma membrane, but also by limiting metabolic activity through effects on the ER, mitochondrial calcium and metabolic flexibility. Although this ER connection is not associated with cell death or impaired growth, I think it is worthwhile addition to the knowledge of B cell physiology and lymphoma pathology.

This manuscript is much improved from the previous version and is now much more convincing in showing the connection of the effects of the immunoglobulin in the ER to metabolism. Although the original work on the Ramos cells was limited by the independent derivation and adaptation of the BCR and SYK KO clones, the new data reproducing the phenotype in DG75 cells and in primary B cells show that the conclusions are valid across cellular models. Most importantly, the experiments with primary cells show that the deletion of the BCR heavy chain reduce ER size independently of rescued survival (by constitutively active PI3K or over-expression of BCL2). Thus expression of the BCR does have effects on the ER that are independent of canonical ITAM signalling.

Two minor points:

In Fig 8 it would be worthwhile to explain in text or legend that the B1-8 deletion is done in in vitro cultures with BAFF.

We have added this information to the figure legend.

I can't find evidence that Ig is required for ER function or XBP splicing in the three references given on line 271: Bakunts, Orsi et al., 2017, Ma, Shimizu et al., 2010, Shaffer, Shapiro-Shelef et al., 2004. I think it is the Iwashima 2003 paper that shows that.

We have replaced the reference.

April 15, 2020

RE: Life Science Alliance Manuscript #LSA-2020-00700-TR

Dr. Julia Jellusova
Albert Ludwigs University Freiburg
Schänzlestraße 18
Baden Württemberg 79104
Germany

Dear Dr. Jellusova,

Thank you for submitting your Research Article entitled "Immunoglobulin expression in the endoplasmic reticulum shapes the metabolic fitness of B lymphocytes". It is a pleasure to let you know that your manuscript is now accepted for publication in Life Science Alliance. Congratulations on this interesting work.

DISTRIBUTION OF MATERIALS:

Again, congratulations on a very nice paper. I hope you found the review process to be constructive and are pleased with how the manuscript was handled editorially. We look forward to future exciting submissions from your lab.

Sincerely,
